# The assassin bug *Pristhesancus plagipennis* produces two distinct venoms in separate gland lumens

Andrew A. Walker [1], Mark L. Mayhew[2], Jiayi Jin[1], Volker Herzig[1], Eivind A.B. Undheim[3], Andy Sombke[4], Bryan G. Fry[5], David J. Meritt[2] & Glenn F. King[1]

The assassin bug venom system plays diverse roles in prey capture, defence and extra-oral digestion, but it is poorly characterised, partly due to its anatomical complexity. Here we demonstrate that this complexity results from numerous adaptations that enable assassin bugs to modulate the composition of their venom in a context-dependent manner. Gland reconstructions from multimodal imaging reveal three distinct venom gland lumens: the anterior main gland (AMG); posterior main gland (PMG); and accessory gland (AG). Transcriptomic and proteomic experiments demonstrate that the AMG and PMG produce and accumulate distinct sets of venom proteins and peptides. PMG venom, which can be elicited by electrostimulation, potently paralyses and kills prey insects. In contrast, AMG venom elicited by harassment does not paralyse prey insects, suggesting a defensive role. Our data suggest that assassin bugs produce offensive and defensive venoms in anatomically distinct glands, an evolutionary adaptation that, to our knowledge, has not been described for any other venomous animal.

---

[1] Institute for Molecular Bioscience, The University of Queensland, St Lucia, QLD 4072, Australia. [2] School of Biological Sciences, The University of Queensland, St Lucia, QLD 4072, Australia. [3] Centre for Advanced Imaging, The University of Queensland, St Lucia, QLD 4072, Australia. [4] Zoological Institute and Museum, Cytology and Evolutionary Biology, University of Greifswald, Soldmannstraße 23, Greifswald 17487, Germany. [5] Venom Evolution Lab, School of Biological Sciences, The University of Queensland, St Lucia, QLD 4072, Australia. Correspondence and requests for materials should be addressed to A.A.W. (email: a.walker@uq.edu.au) or to G.F.K. (email: glenn.king@imb.uq.edu.au)

Animals produce venoms to enhance their success in diverse ecological interactions, including hunting, defence, parasitism and intraspecific competition[1]. While some venoms are used for highly specific purposes—such as mating competition in platypuses[2] or endoparasitism by ichneumonid wasps[3]—others are used for multiple purposes[4]. Multifunctional use of venom is possible due to the presence of numerous different molecules in venom, and their multifaceted pharmacological effects (e.g., activators of voltage-gated sodium channels may cause either paralysis during prey capture, or pain during defence[5]). Nevertheless, some venoms will be better suited to certain interactions than others depending on their composition, which might result in selection for the ability to vary venom composition. Some animals, such as the marine cone snails *Conus geographus* and *Conus marmoreus*, can inject distinct predaceous and defensive venoms depending on external stimuli[6]. The scorpion *Parabuthus transvaalicus* has been reported to inject a 'pre-venom' rich in potassium ions in defensive encounters, whereas a more peptide-rich venom is used for prey capture[7]. In cnidarians such as sea anemones, toxin delivery systems are particularly complex and comprise large numbers of individual nematocysts and gland cells across the surface of the animal. Although it is likely that venom delivered by feeding tentacles differs from venom delivered by tissues specialised for defence or intraspecific competition[8–10], this has not been conclusively demonstrated.

Assassin bugs are the predatory, venomous insects in family Reduviidae (Insecta: Hemiptera), excluding the kissing bugs (subfamily Triatominae) that feed on vertebrate blood. Reduviidae is part of suborder Heteroptera, a group that diverged from other hemipteran insects such as cicadas and aphids ~250 mya[11], co-incident with a trophic shift from phytophagy to predation[12]. The feeding and salivary system of their phytophagous forebears, comprising complex labial glands and piercing-sucking mouthparts, was adapted by heteropterans to form a venom apparatus. The overall arrangement of the glandular structures nevertheless remains similar: hemipteran labial glands are paired structures in the thorax and abdomen, comprising the anterior (AMG) and posterior lobes (PMG) of the main gland plus an accessory gland (AG)[12]. The two lobes of the main gland, as well as the venom duct (VD) and AG duct, converge at a structure called the hilus, which has been reported to consist of several 'mixing chambers' separated by muscle-controlled valves[13,14]. In non-venomous hemipterans, such as the phytophagous bug *Oncopeltus fasciatus* (Lygaeidae), this anatomical arrangement allows functional specialisation of each gland compartment, with each compartment contributing differently to the protein-rich 'salivary sheath' material and digestive saliva[15]. Predaceous heteropterans, including predatory stinkbugs (Asopinae)[16] and giant water bugs (Belostomatidae)[17], have also been reported to produce distinct secretions in each gland region. Extracts of the AMG from the assassin bugs *Haematorrhophus nigroviolaceus* and *Peirates affinis* were shown to induce rapid paralysis of prey, whereas PMG extracts caused no immediate effects but resulted in death after several hours[14]. Thus, the reduviid AMG and PMG are suggested to be specialised for secretion of neurotoxins and digestive enzymes, respectively[18]. However, other authors have found less marked or no difference between the bioactivities of AMG and PMG gland extracts[19–21]. We recently reported that venom obtained by electrostimulation of the assassin bug *Pristhesancus plagipennis* is a complex mixture of catabolic enzymes, putative cytolytic/pore-forming proteins and putatively neurotoxic disulphide-rich peptides[22]. However, neither the gland compartment that produces this venom nor the biological role of each compartment is known.

Here we address these issues using complementary imaging technologies, gland compartment-specific transcriptomics and proteomics, as well as toxicological assays. We report that the AMG and PMG of *P. plagipennis* produce distinct, complex venoms that can be elicited with different stimuli. Our data highlight the novelty and complexity of assassin bug venom systems and provide evidence for the functional convergence of venom use between assassin bugs, cone snails, scorpions and cnidarians. However, in contrast to these other groups, assassin bugs possess a set of anatomical structures clearly suited to the delivery of multiple complex venoms, as we demonstrate in this study.

## Results

**Visualisation of venom glands.** In order to visualise the assassin bug venom apparatus, we performed three-dimensional (3D) reconstructions using magnetic resonance imaging (MRI), micro-computed tomography (µCT), and confocal laser scanning microscopy (CLSM). MRI revealed the extent of the venom glands and their position within the body of an intact adult female (Fig. 1a–h). In this specimen, the paired main glands measured 9.6 mm long and extended from the anterior thorax into the abdomen. The teardrop-shaped lumen of the AMG is clearly separate to that of the much larger PMG (Fig. 1e). The volumes of AMG, PMG and AG lumens in the individual imaged by MRI were 0.32, 5.21 and 1.36 mm³, respectively. We conclude that the venom glands of *P. plagipennis* are compartmentalised into separate lumens of the AMG, PMG and AG.

The hilus (H) connecting the AMG and PMG consists of an inner and outer chamber (HI and HO, respectively; Fig. 1f–h; Supplementary Movie 1) similar to previously investigated reduviids[23,24]. Visualisation of the hilus stained with 4′,6-diamidino-2-phenylindole (DAPI) and phalloidin, which stain cell nuclei and F-actin, respectively, reveals a network of muscle fibres (M) extending over the surface of PMG and AMG (Fig. 1i). At the point where each lobe joins the hilus, this muscle network converges to form a sphincter valve (S). An antibody against acetylated tubulin strongly stained structures with the characteristic appearance of nerve processes. These processes, supported by trachea attached to the VD, extensively innervate muscle fibres close to the hilus and on the surface of the glands (Fig. 1j, Supplementary Figure 1) revealing a potential for controlling venom outflow as summarised in Fig. 1k. The VD extends anteriorly from the hilus to the venom pump (VP) of the head. The VP is a hollow, conical structure with a large muscle bundle connected to the centre of the concavity (Fig. 1l, m), an arrangement noted in other hemipterans to resemble a piston[25,26]: muscle contraction will fill the pump cavity, while relaxation of the bundle will cause expulsion of the venom through the venom channel (VC) that extends through the ventral part of the proboscis and ultimately into the stylet tips. We did not observe a 'complex valve' as exists at the junction of the VP and VC in giant water bugs (Belostomatidae)[27], suggesting its absence in reduviids.

Our imaging data show that the assassin bug venom apparatus comprises numerous anatomical structures with the potential to modulate venom composition: three anatomically separate gland lumens; networks of muscle fibres surrounding each lobe of the main gland; muscular sphincter valves between each lobe of the main gland and the hilus, as well as the VP of the head.

**The AMG and PMG secrete different proteomes.** To examine the secretory output of each gland compartment we utilised transcriptomes of the PMG, AMG and AG[22]. Each of these transcriptomes represents a single sequencing experiment

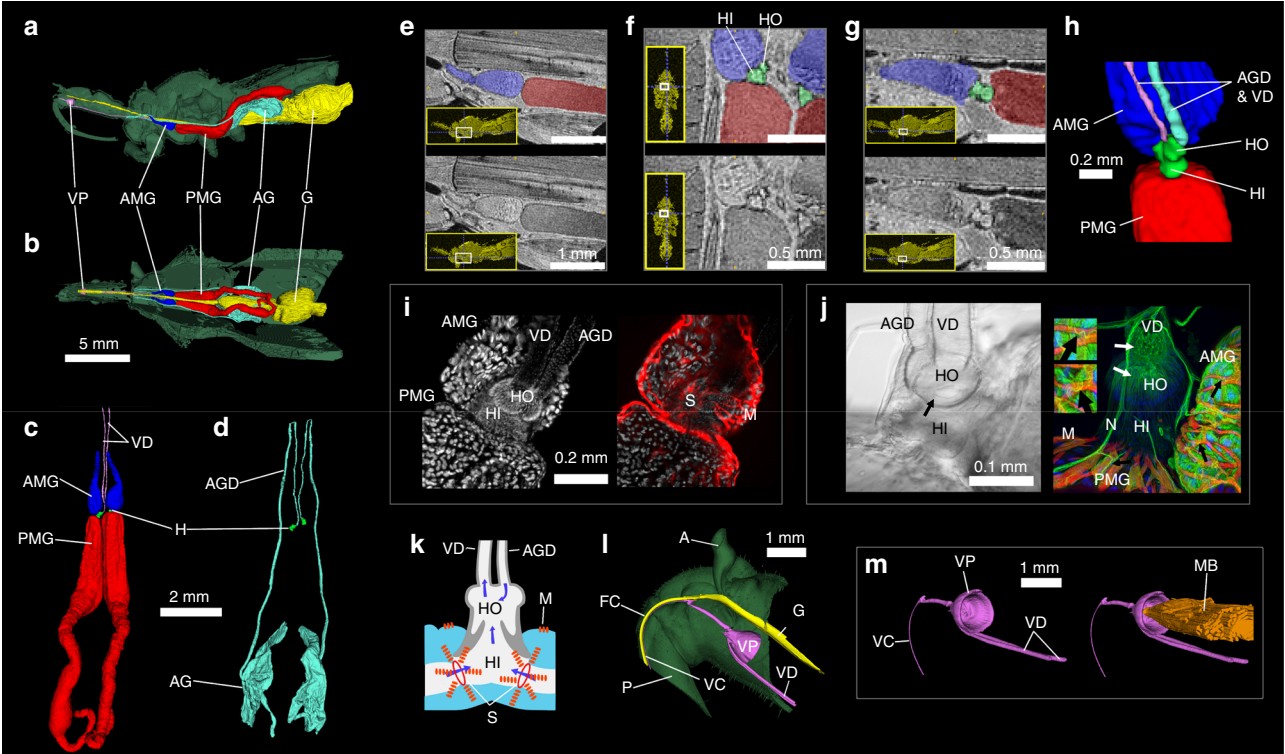

**Fig. 1** Assassin bug venom apparatus. **a–d** Gland reconstructions from MRI. **a** Sagittal view and **b** dorsal view, showing position of glands in relation to the surface of the insect (green). VP, venom pump (pink); AMG, anterior main gland (dark blue); PMG, posterior main gland (red); AG, accessory gland (light blue); G, gut (yellow). **c** Dorsal view of main gland. VD, venom ducts connecting the hilus (H, green) to VP (pink). **d** Dorsal view of AG and accessory gland ducts (AGDs). **e–g** 2D MRI slices with compartments in upper panels coloured dark blue (AMG), red (PMG) and green (hilus). Insets show location of each slice. **e** Parasagittal slice showing separate AMG and PMG lumens. **f** Dorsal plane slice showing inner (HI) and outer (HO) chambers of hilus. **g** Parasagittal slice (more medial than that in **e**), showing hilus connecting AMG and PMG. **h** MRI reconstruction of hilus showing connectivity to AMG, PMG, VD and AGD. **i** Confocal laser scanning microscopy (CLSM) images of the hilus and surrounding structures. Left, nuclear stain showing arrangement of hilus, AMG, PMG, VD and AGD. Right, nuclear stain (grey) with actin stain (red) superimposed. Concentrated actin in muscle fibres (M) reveals a sphincter valve (S) at the junction of AMG and HI. **j** CLSM and differential interference contrast (DIC) images of hilus region. Left, DIC image. Right, antibody against acetylated tubulin (green) highlights nerves (N) and neuromuscular junctions (black arrows) as well as fibrous-appearing staining near the start of VD (white arrows) and gland cells. Red: actin staining of muscle fibres. Blue: nuclear stain. Insets: enlargements of neuromuscular junctions marked by black arrows. **k** Schematic of hilus with arrows indicating putative direction of flow of gland contents. Red bands represent muscle fibres (M). **l–m** μCT scans showing arrangement of VP and VD in relation to gut (yellow). **l** Sagittal view of the head. VC, venom channel; FC, food channel; P, proboscis; A, antenna. **m** Oblique posterolateral view showing strong concavity on the posterior surface of the VP (left) connected to muscle bundle (MB, orange)

conducted on mRNA from a specific gland compartment, using mRNA of glandular tissue pooled from four individuals. To produce putative 'secretomes' (i.e., lists of the putative secreted proteins in each compartment), we filtered transcriptomes for contigs encoding open reading frames (ORFs) with a high-confidence secretion signal peptide, putative coding region >150 bp and stop codon, but without endoplasmic reticulum (ER) retention signal or predicted transmembrane regions (see Methods). Clustering the results from each gland compartment yielded a total of 479 predicted secreted proteins transcribed in the venom gland complex. Transcript abundance (fragments per kilobase million, FPKM) was quantified by mapping the trimmed reads from each compartment to the appropriate transcriptome. Transcripts encoding putative secreted proteins accounted for a large amount of reads from the AMG (49%) and PMG (50%) pools but not from the AG pool (3%; Fig. 2a). These data suggest that the AMG and PMG are the main secretory tissues of the venom glands, which is consistent with the high yield of polyA(+) mRNA obtained from the AMG and PMG pools (356 and 226 ng per mg tissue, respectively) but not the AG pool (13 ng per mg tissue) obtained during library preparation[22]. Each gland compartment was observed to produce a highly divergent set of

transcripts, with FPKM towards shared contigs accounting for <3% of the overlap of AMG and PMG (Fig. 2b).

To further investigate secretory output of each gland compartment, we annotated each putative secreted sequence using HMMER and BLAST searches, and proteins were classified into groups based on homology or inferred function (see Methods; Supplementary Data 1). At the protein family level, the most abundantly encoded transcripts in the PMG (Fig. 3a) are members of the pore-forming trialysin/redulysin toxin family (total FPKM 67,488), followed by proteases (42,429), venom family 1 proteins (8326), cystatins (6513), CUB domain proteins (3076) and peptides homologous to the putatively neurotoxic Ptu1 peptide from *Peirates turpis* (2626). Overall, the major protein classes expressed in the PMG closely mirror those identified in venom obtained by electrostimulation[22]. In contrast, the most abundant transcripts in the AMG encode two novel proteins (secretome IDs #28 and #437 in sheet A of Supplementary Data 1; combined FPKM 55,406), members of the 'hemolysin-like' protein family[28,29] (combined FPKM 36,712), cystatins (11,251), other peptides (4207), Kazal domain peptides (3091) and Ptu1 family peptides (1265). To test these trends statistically over the entire secretomes at the level of protein

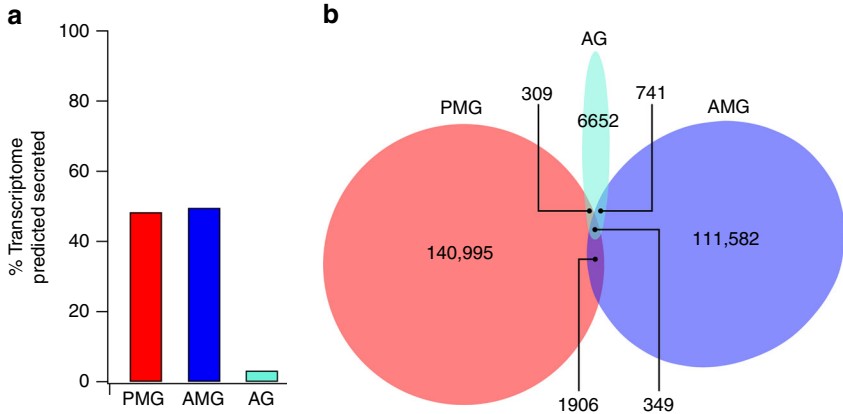

**Fig. 2** Predicted secreted proteins in each compartment of the venom glands. **a** Secretory activity, as measured by the proportion of total transcription encoding predicted secreted proteins (normalised to fragments per kilobase million, FPKM). PMG, main gland posterior lobe; AMG, main gland anterior lobe; AG, accessory gland. **b** Overlap of transcriptional activity between gland compartments. Each area represents transcript abundance (FPKM) of all secretome entries that are separate or shared between the three compartments

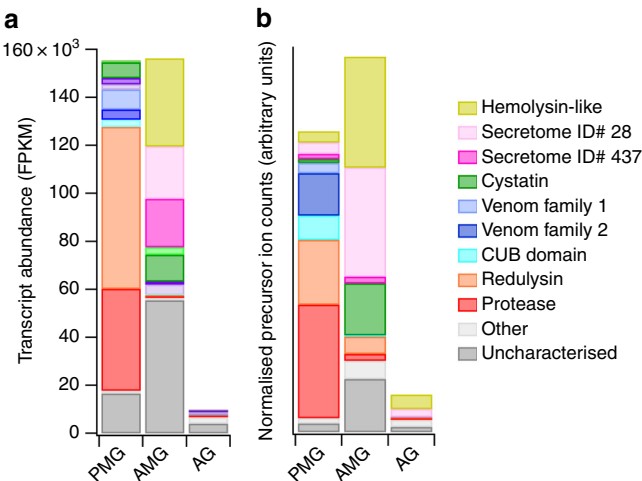

**Fig. 3** Different classes of protein are secreted in each gland compartment. **a** Transcript abundance of major classes of secreted protein classes in each gland compartment. **b** Proportion of precursor ion counts observed by LC-MS/MS originating from each major protein class (putative housekeeping proteins excluded). PMG, posterior lobe of main gland; AMG, anterior lobe of main gland; AG, accessory gland

annotated using the HMMER, BLAST, SignalP and TMHMM algorithms (see sheets A-D in Supplementary Data 2; see Methods). To focus on possible venom proteins, we excluded sequences likely to represent non-venom 'housekeeping' proteins (e.g., actin and ribosomal proteins; see Methods) from further quantitative analysis. A total of 45, 51 and 12 possible venom proteins were identified from the PMG, AMG and AG extract, respectively. For the PMG and AMG lobes, the proportion of precursor counts arising from each protein class (Fig. 3b) closely matched transcript abundances observed previously (Fig. 3a). In the PMG, the three most abundant protein classes represented were proteases (38%), redulysins (21%), venom family 2 proteins (14%) and CUB domain proteins (8%). In the AMG, hemolysin-like proteins (29%), cystatins (14%) and unknown proteins (39%) were the three most abundant classes. Of all non-housekeeping precursor counts detected from the AMG, 29% originated from a single protein in the unknown class, which is also the product of the highest-abundance transcript in the AMG secretome (secretome ID #28). Results were similar in the remaining two individuals analysed: proteases and redulysins were more confidently identified than hemolysin-like proteins in the PMG, whereas the reverse was true in the AMG (see sheets E-M in Supplementary Data 2). These results demonstrate that the PMG and AMG each secrete a different set of proteins and peptides, which then accumulate within their respective gland lumens.

family, we compared FPKM values for redulysins, CUB domain proteins, proteases, hemolysin-like proteins, Kazal domain proteins and Ptu1 family proteins using the Kruskal–Wallis test (see sheet B of Supplementary Data 1). Proteases ($p < 10^{-9}$) and redulysins ($p = 0.0013$) were preferentially expressed in the PMG, whereas hemolysin-like proteins were preferentially expressed in the AMG ($p = 0.005$). Of the Ptu1 family peptides, Pp4 was more highly expressed in the AMG, whereas Pp1−3 and Pp5 were more highly expressed in the PMG (Supplementary Figure 2).

We investigated if differences observed in transcript abundance between compartments in RNA-sequencing (RNA-Seq) experiments corresponded to differences in the abundances of proteins accumulated in each gland lumen. The PMG, AMG and AG were extracted from three adult bugs and the protein content of the nine extracts analysed individually using liquid chromatography coupled tandem mass spectrometry (LC-MS/MS). For the first individual, mass spectra were used to identify proteins against all transcriptomic data, and protein abundances were quantified using precursor spectral counts. Each identified protein was also

**Harassment and electrostimulation yield distinct venoms.** Venom secreted by the AMG is not abundant in venom harvested by electrostimulation. For example, for the 34 most highly expressed secretome contigs in the AMG (FPKM 100–21,859), most protein products (24 proteins or 71%) were not detected in venom harvested by electrostimulation[22]. We therefore investigated if a different extraction method or stimulus could induce *P. plagipennis* to eject the contents of the AMG through the proboscis. We noted that awake, restrained bugs sometimes deposit a much smaller quantity of venom (0.2−1 µl, as opposed to 5−10 µl after electrostimulation) in response to the proboscis being placed into a collecting tube (Supplementary Movie 2). Gentle harassment with tweezers evokes this small quantity of venom more reliably (Supplementary Movie 3). In unrestrained bugs, the production of small amounts of venom from the proboscis prior to electrostimulation is associated with characteristic defensive posturing (Supplementary Movie 4). To examine the content of venom obtained by harassment, we obtained eight samples by this method from different individuals and examined

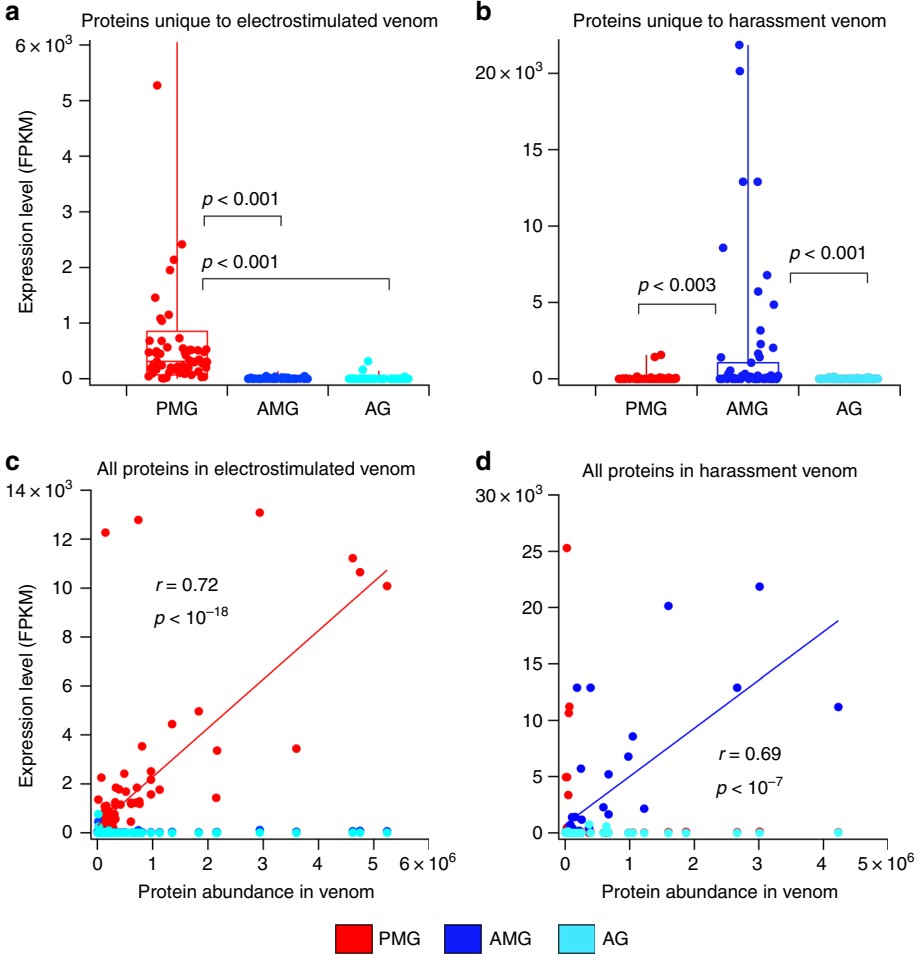

**Fig. 4** Electrostimulation yields PMG venom and harassment yields AMG venom. **a**, **b** Expression levels of proteins unique to venoms obtained either by electrostimulation (**a**) or harassment (**b**). **c**, **d** Comparison of protein abundance (calculated from precursor ion counts) and expression level in each gland compartment. **c** Venom obtained by electrostimulation. Values of $r$ and $p$ indicate the regression to PMG expression values. **d** Venom obtained by harassment. Values of $r$ and $p$ are for regression to AMG expression values

the protein content using LC-MS/MS. A pooled Paragon search of mass spectra from all eight samples (against ORFs from all gland compartments) yielded a total of 150 proteins, of which 72 were not previously reported in *P. plagipennis* venom[22]. Identified protein sequences were annotated by HMMER and BLAST searches (see sheet A in Supplementary Data 3; see Methods). Newly detected proteins included hemolysin-like proteins (10), Kazal domain peptides (5) and a CRiSP/Antigen-5 family protein (1). Of the 24 'missing' proteins that are encoded by transcripts with high abundance in the AMG (FKPM 100–21,859) but were not previously detected in venom obtained by electrostimulation, 22 (92%) were detected in venom obtained by harassment. A total of 33 proteins and 14 peptides from novel or uncharacterised families, including short disulphide-rich peptides with novel cysteine frameworks such as Pp18a (STIPNNICQTCCVNPSS-DPRCASVRCNCPIKTSPPCSE) and Pp19a (VCWDTGCQLNA-WAVRGCAQYGMRDVNMKSCSGGIIYTCCD), were detected. We further examined the contribution of between-individual variability to this result by performing separate Paragon searches for each of eight samples of venom obtained by harassment and six samples obtained by electrostimulation. All eight venom samples obtained by harassment showed similar profiles in which hemolysin-like proteins were identified with higher confidence than proteases or redulysins (see sheets C-J in Supplementary Data 3). The reverse was true for six samples obtained by

electrostimulation (see sheets K-P in Supplementary Data 3). Thus, samples collected from *P. plagipennis* differ markedly depending on the method by which they are collected.

To quantitatively investigate the glandular source of venom collected using either electrostimulation or harassment, we compared the expression levels of 'marker' proteins (i.e., those detected only in one type of venom) between each of PMG and AMG. Proteins unique to venom obtained by electrostimulation were more highly expressed in PMG and showed weak expression in other compartments (Fig. 4a; $p < 0.001$; see sheet A in Supplementary Data 4). In contrast, proteins that were only detected in venom obtained by harassment were more highly expressed in AMG (Fig. 4b; $p < 0.001$; see sheet B in Supplementary Data 4). To further investigate the connection between the site of expression and abundance in venom collected by either manner, we quantified the proteins detected in a representative sample of venom, obtained by either harassment or electrostimulation, from observed precursor ion counts. For venom collected by electrostimulation, protein abundance correlated strongly ($r = 0.72$; $p < 10^{-18}$) with expression levels in the PMG but not in the AMG (Fig. 4c; see sheet C in Supplementary Data 4). Likewise, transcript abundance in the AMG (but not in other gland regions) is a strong predictor of protein abundance in venom obtained by harassment ($r = 0.69$, $p < 10^{-7}$; Fig. 4d; see sheet D in Supplementary Data 4). Thus,

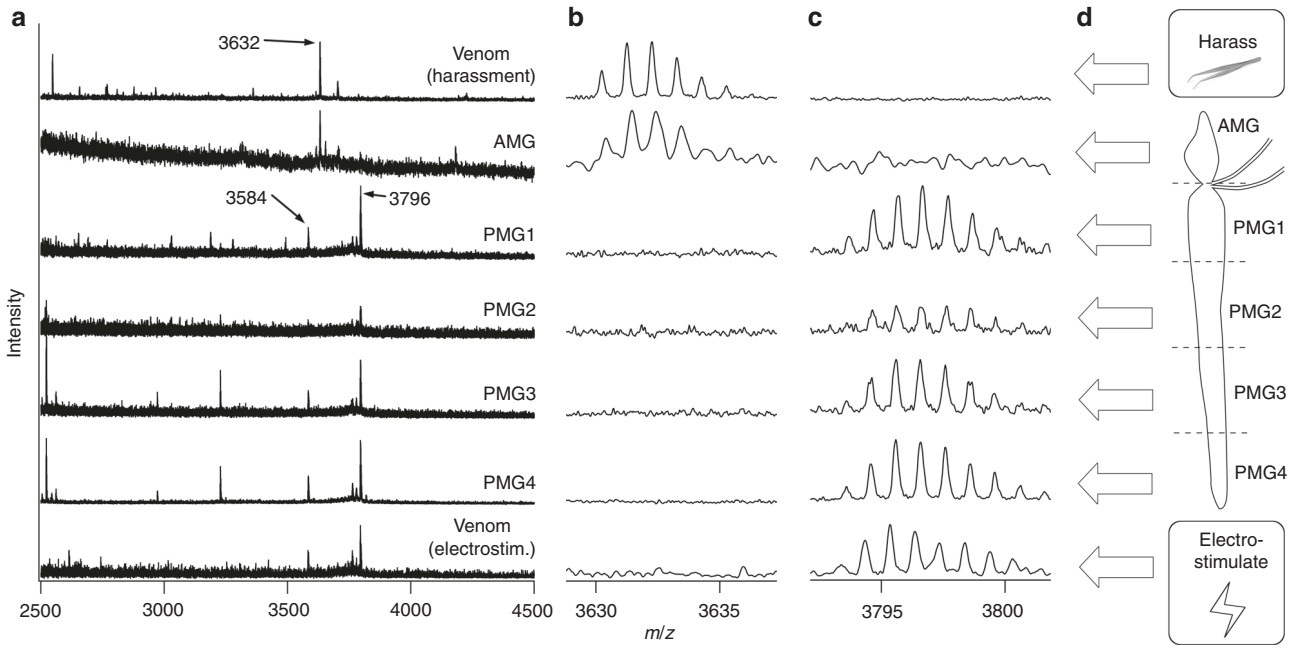

**Fig. 5** MALDI spectra of venoms and venom gland extracts. Each spectrum was obtained using MALDI-TOF of samples obtained from the same adult male bug. **a** MS spectra acquired over the range m/z 2500–4500 Da. Expansions of the regions around 3632 Da (**b**) and 3796 Da (**c**) reveal close correspondence of detected masses in these regions between venom samples and gland extracts. **d** Schematic showing the source of samples analysed. Samples PMG1–PMG4, posterior lobe extracts; AMG, anterior lobe extract

venom secreted by, and stored within, the AMG can be elicited by the harassment of bugs. Venom obtained by electrostimulation is secreted and stored in the PMG.

To further confirm this finding, we analysed venom and gland extracts using matrix-assisted laser desorption-ionisation-time of flight (MALDI-TOF) mass spectrometry. MALDI-TOF mass spectra of six individual venom samples obtained by electrostimulation were similar (Supplementary Figure 3a), whereas eight individual samples obtained by harassment were more variable, despite all experimental animals being siblings (Supplementary Figure 3b). For this reason, we compared MALDI spectra of samples obtained over a 2-week period from one individual (an adult male) comprising venoms obtained by both harassment and electrostimulation and gland extracts from the AMG and the PMG (Fig. 5). A close correspondence was observed between venom obtained by electrostimulation and the contents of the posterior lobe, with both spectra dominated by strong signals at m/z 3796 and 3584 Da, which correspond to the Ptu1 family peptides Pp1a and Pp3, respectively[22]. Extracts of different portions of the elongated posterior gland (PMG1–PMG4) yielded similar mass spectra. In contrast, the mass spectrum of venom obtained by harassment closely matched the extract of AMG, with a strong signal at m/z 3632 but lacking signal at m/z 3796 or 3584 Da (Fig. 5).

**Venom from the PMG but not from the AMG prevents prey escape**. To gain insights into the biological function of each gland, we injected AMG ($n = 5$) and PMG ($n = 5$) extracts into blowflies (*Lucilia cuprina*), using doses of 1.7 μl (equivalent to 5.6% of the paired venom gland compartments from a single individual). AMG extract had no effect at 15 or 60 min, whereas injection of PMG extract produced 100% paralysis and 60% death after 15 min, and 100% mortality after 60 min. To clarify if such toxicity might facilitate prey capture, we tested venoms obtained by harassment or electrostimulation in an escape assay using juvenile crickets. In this assay, the time taken for third-instar crickets

(~50 mg) to escape an upturned Petri dish lid was measured immediately after injection of venom, up to a maximum of 300 s. Crickets injected with 1.7 μl water escaped quickly (5.5 ± 7.2 s; mean ± SD, $n = 5$). In contrast, injection of venom obtained by electrostimulation resulted in immediate and marked motor effects, including loss of coordinated movement, paralysis and antenna twitching (Supplementary Movie 5). These effects closely recapitulate the effects of envenomation by freely hunting *P. plagipennis*[22], resulting in a complete loss of effective escape behaviour ($p < 0.001$; Fig. 6a). All crickets injected with 0.17, 0.34 or 1.7 μl of venom obtained by electrostimulation failed to escape the dish at all, dying 5–15 min later. Injections of more dilute venom revealed that a very small amount of PMG venom is required to prevent escape in this assay venom ($ED_{50} = 0.079$ μl; Fig. 6b). In contrast, injection of venom obtained by harassment at levels that cause immediate paralysis (0.17 μl) had mild effects on escape behaviour. Of three crickets injected with this dose only one displayed mild motor effects, and all crickets escaped, taking an average time of 37 ± 5 s. Thus, in contrast to a previous report[23], our data indicate that the PMG, not the AMG, is the source of venom used by *P. plagipennis* for prey capture.

## Discussion

In this study we demonstrated that the assassin bug *P. plagipennis* accumulates two distinct venoms in two separate secretory gland lumens, and that these two venoms can be elicited by different stimuli. Thus, the assassin bug venom system has functional similarities to those of cone snails and scorpions that are also capable of injecting multiple venoms[6,7]. However, the physiological mechanisms allowing such behaviour in other taxa are poorly described. In cone snails, secretory cells lining the distal and proximal parts of the VD secrete toxins specialised for either defence or predation into the VD. Since the same duct is shared by both regions, it is hypothesised that toxin secretion occurs shortly before envenomation in response to descending nervous control[6]. In comparison, assassin bugs possess multiple

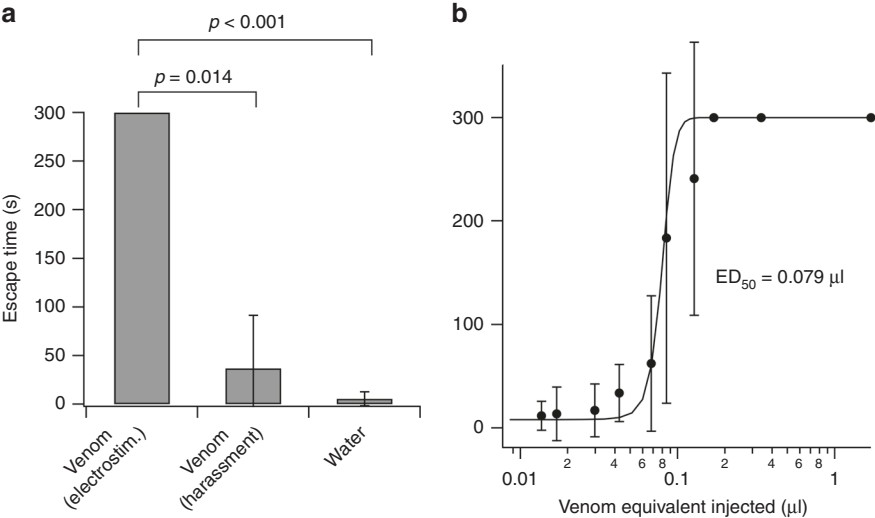

**Fig. 6** Venom obtained by electrostimulation but not by harassment paralyses insects. **a** Effect of injecting venom obtained by electrostimulation or harassment, or water, on cricket escape. For each venom condition, 0.17 µl venom equivalent was injected into the abdomen and the time to escape an upturned Petri dish lid (in s, up to 300 s, mean ± SD) was scored. **b** Dose–response curve for inhibition of escape success by venom obtained by electrostimulation

anatomical structures sufficient to allow the injection of multiple venoms: separate gland lumens that accumulate distinct sets of venom proteins; sphincter valves at the opening of each gland; muscle networks surrounding glands to expel their contents; the VP between the VD and the proboscis; and nerve pathways descending from the central nervous system (CNS) to control these. The details of how each component of the venom system functions during envenomation remain to be elucidated, but most likely the glandular valves, muscle networks and VP are activated by CNS input in a coordinated manner to inject the appropriate venom. These results also explain why harvesting venom using electrostimulation is not always successful or instantaneous, and why it does not expel the contents of all venom gland compartments, since electrically induced muscle contraction might either restrict or facilitate venom flow at multiple points.

Both the AMG and PMG are highly active secretory glands that secrete distinct venoms into separate gland lumens. In contrast, secretory activity in the AG is low. Thus, although the AG transcribes numerous proteins and peptides that might be assessed as 'toxin-like' by homology only (e.g., defensins and mys2), our results support the traditional view that the heteropteran AG serves primarily as a water reservoir[14,18,19]. Assassin bugs such as *P. plagipennis* use venom in at least three different ways: prey capture; defence; and extra-oral digestion (EOD). Of these, EOD is most likely to be performed by PMG venom, which is rich in proteases. We found that PMG venom caused immediate paralysis followed by death when injected into crickets, closely recapitulating the observed effects of natural envenomation. The amount of venom required for this effect (ED$_{50}$ = 0.079 µl) suggests that adult bugs, which have PMG lumens of >5 mm$^2$ and frequently produce 5–10 µl venom during electrostimulation, carry enough venom to paralyse 100–200 crickets of the size tested (~50 mg). PMG venom is therefore likely used for predation, in which it facilitates both prey capture and EOD, as originally hypothesised[22]. In contrast, the AMG venom produced only mild or no effects when injected in similar doses, and it is elicited when the bugs are threatened, suggesting a defensive role. However, our results do not rule out alternative hypotheses. For example, the AMG might secrete proteins and peptides that act synergistically with PMG-secreted toxins during predaceous envenomation, or produce toxins that prevent haemolymph

clotting. Currently, we are conducting experiments to clarify the role of each of the venoms and their individual toxins.

Our results contrast with those of Haridass and Ananthakrishnan[14] who injected extracts of gland compartments into prey animals, and concluded that the AMG extract contained neurotoxins while the PMG extract contained enzymes for EOD. However, other studies have found fewer functional differences between the glands using similar assays[20,21], and Edwards[19] found that both PMG and AMG secretions potently paralysed cockroach heart-dorsum preparations. Apart from different assay conditions, phylogeny may explain some of these differences, as Haridass and Ananthakrishnan studied more basal ectrichodiine and peiratine assassin bugs, whereas our study and the others cited investigated more derived reduviine and harpactorine bugs. Interestingly, we note that in two cimicomorphan species representing lineages that have independently transitioned from predaceous insectivory to parasitic haematophagy—the kissing bug *Rhodnius prolixus* (reduviid subfamily Triatominae) and the bed bug *Cimex lectularius* (Cimicidae)—the main venom gland has been reduced to a single lobe[13]. These observations are consistent with the apparent reduction or lack of either digestive or defensive venom use by these insects. Both PMG and AMG lobes have been retained in another the blood-feeding triatomine genus, *Triatoma*[13]. However, we are not aware of any study comparing the secretory output of the two gland lobes in this genus.

This study clarifies the physiological division of labour in a complex anatomical structure, the assassin bug venom gland. In addition, it increases the number of known assassin bug toxins, and provides new insights into the evolution of numerous gene and protein families in Heteroptera. The composition of harassment/AMG venom demonstrates that proteins from the Kazal, hemolysin-like, CRiSP and pacifastin families, among others, have been recruited into the venom of insectivorous heteropterans. Moreover, venom obtained by harassment was found to contain 14 families of venom proteins and 13 families of venom peptides that are novel or uncharacterised, in addition to those we have previously reported[22]. Toxins such as Kazal domain peptides —which have also been detected in the venom of blood-feeding reduviids[30]—were likely recruited into venom prior to the shift from insectivory to haematophagy. A limitation to our study is

that we have not analysed differences in venom composition between males and females, or between the different immature and adult stages, or between virgin and mated individuals. Additional studies may identify further venom peptides and proteins, and patterns of venom variation, in this species.

Our results highlight the sophistication of the assassin bug venom system, and demonstrate the importance of combining multiple experimental techniques to characterise complex venom systems in novel taxa. Moreover, clarification of the functional role of the AMG and PMG has important implications for bioprospecting studies; PMG-derived toxins are likely to be of interest to researchers interested in venoms as a potential source of drugs[31] and insecticides[32], whereas toxins derived from the AMG might be useful for elucidating novel nociceptive pathways[33].

## Methods

**Insects and venom collection.** Assassin bugs (*Pristhesancus plagipennis*) were collected in Brisbane, Australia, and fed on crickets (*Acheta domesticus*; Pisces Enterprises, Kenmore, QLD, Australia). Both wild-caught and captive-bred animals were used in this study. Species-level identification was performed according to a published key[34] and two voucher specimens were deposited in the Queensland Museum Entomology Collection with reference numbers T239616 (male) and T239617 (female). Venom was harvested from adult bugs more than 1 week after their final moult, and 4–6 days after feeding. Venom harvest was by electro-stimulation[22] or by gently harassing awake but restrained bugs by touching legs, antennae and abdomen with a pair of tweezers to simulate attack by a small animal. Venom was immediately transferred to a tube on ice and stored at −20 °C until analysis. For toxicological assays, venom was freeze-dried and reconstituted at 100 mg/ml (estimated from $A_{280}$) in water. Gland protein extracts were prepared by placing dissected tissue in 50 μl of phosphate-buffered saline (PBS) on ice, vortexing for 10 s, centrifuging for 1 min at 5000 rcf at 4 °C to release the contents of the gland lumen and removing tissue with tweezers. The sample was then clarified (10 min, 17,000 rcf, 4 °C) and the supernatant analysed.

**Gland imaging and immunostaining.** For gland imaging, adult bugs were anaesthetised with $CO_2$ and fixed in either neutral buffered formalin (MRI) or Bouin's fixative (μCT). For MRI, Magnevist® (gadopentetate dimeglumine; Bayer) was added to enhance contrast and the bugs imaged on a 16.4T Bruker Avance MRI spectrometer at a resolution of 30 μm. For μCT, fixed specimens were dehydrated using a graded ethanol series and incubated overnight in 1% alcoholic iodine solution (Carl Roth #X864.1). After washing in 99.8% ethanol, the specimens were critical point dried using an automated dryer Leica EM CPD300 (Leica Microsystems)[35] and mounted on an insect pin with a hot glue gun. Scanning was performed using a XRadia MicroXCT-200 X-ray imaging system (Carl Zeiss) equipped with switchable scintillator-objective lens units, which gives a flexible field of view[35]. Tomographies were performed at ×10 magnification, exposure times of 2 s and calculated pixel sizes of 1.96 μm. X-ray source settings were 40 kV and 8 W resulting in 200 mA source current. Tomography projections were reconstructed using XMReconstructor software (Carl Zeiss) yielding image stacks in TIFF format. All scans were performed using Binning 2 (summarising 4 pixels, resulting in noise reduction) and subsequently reconstructed using Binning 1 (full resolution) to avoid information loss. For both MRI and μCT data, anatomical structures were then visualised by manual and automated segmentation and 3D visualisation using ITK-Snap 3.2.0[36].

For fluorescence microscopy of the hilus, venom glands were dissected intact in PBS then fixed in 3.7% formaldehyde in PBS for 30 min, followed by three washes in PBS for 10 min each at room temperature. Specimens were blocked in a solution of PBS containing 0.4% Triton X-100 (PBT) and 2% normal goat serum (NGS) for 1 h at room temperature then incubated overnight at 4 °C in mouse anti-acetylated tubulin antibody (Sigma) at 1:200 dilution in PBT/2% NGS. This was followed by three washes in PBT for 10 min each prior to incubation in Alexa Fluor® 488 anti-mouse secondary antibody (Invitrogen) at 1:100 dilution, Alexa Fluor® 568 phalloidin (Invitrogen) at 1:50 dilution and 1 mg/ml DAPI in PBT/2% NGS for 4 h at room temperature. Specimens were washed three times in PBT for 10 min each then placed in 70% glycerol in PBS prior to mounting. Differential interference contrast microscopy was conducted using a Zeiss Axioskop compound microscope and images were captured by a Toupcam colour digital camera and Touplite software. CLSM was performed on an Olympus FV1000 microscope supported by Olympus FV10-ASW software.

**RNA-Seq and transcriptomics.** Transcriptomes of the AMG, PMG and AG were generated from four individuals (two adult females and two last-instar nymphs) by pooling gland compartments between individuals[22]. For RNA extraction, bugs were anaesthetised with $CO_2$ for ~5 min, the venom glands removed and divided into AMG, PMG and AG, and stored in >10× the glandular volume of RNAlater

(Ambion) in separate tubes. After total RNA extraction using a DNeasy kit (Qiagen), mRNA was isolated using a Dynabeads mRNA Direct kit (Ambion) according to the manufacturer's instructions. This process yielded 3120, 680 and 119 ng of mRNA from the PMG lobe, the AMG lobe and AG tissues, respectively. RNA-Seq was performed on 340 ng of AMG and PMG mRNA and 119 ng of AG mRNA on an Illumina NextSeq instrument at the IMB Sequencing Facility. After TruSeq library preparation, each sample was run on four lanes of a 150 cycle mid-output run to generate 2×75-bp paired-end reads (AMG, 78,592,255 reads; PMG, 95,700,337 reads; and AG, 54,480,493 reads). For each of PMG, AMG and AG, eight assemblies were constructed using CLC Genomics Workbench (CLC Bio) and Trinity[37]. For CLC assemblies, reads below 30 were trimmed and the reads assembled using minimum contig length of 150 bp, minimum similarity to join contig 0.95, and word (*k*-mer) sizes of 21, 24, 29, 34, 44, 54 and 64. For Trinity, which employs a fixed *k*-mer method, reads were assembled using the default trimming parameters and minimum contig length of 150 bp. For each gland compartment, contigs from the Trinity and CLC assemblies were pooled and clustered using CD-HIT[38] (threshold 95%) and then re-imported into CLC Genomics Workbench where trimmed reads were re-mapped and used to update the final contigs. For protein identification experiments using LC-MS/MS, a sequence library was constructed by translating all ORFs of 90 bp and greater from the final transcriptomes of all compartments. This yielded 149,776 amino-acid sequences, to which 155 common LC-MS/MS contaminant sequences were added to produce the final sequence database for Paragon searches.

To generate secretomes of contigs encoding putative secreted contigs, the transcriptome of each gland compartment was analysed separately. ORFs larger than 90 bp were examined using the SignalP 4.0 algorithm[39]. ORFs encoding proteins with start methionine, stop codon and SignalP *D*-score > 0.7 were selected. We further eliminated ORFs that partially overlapped other ORFs with BLAST homology to sequences encoding known proteins or non-coding RNA ($E < 0.01$), ORFs encoding proteins with C-terminal KDEL and HDEL endoplasmic retention signals, ORFs with internal transmembrane regions predicted by TMHMM[40], and proteins with BLASTp homology ($E < 0.01$) to insect cuticle proteins and collagen. Each contig was manually reviewed for frameshift errors. Each trimmed sequencing read[22] was then mapped against the final contigs using CLC Genomics Workbench's RNA-Seq analysis tool with similarity cut-off 0.9 and maximum number of hits = 3 before conversion to FPKM values.

**Sequence annotation.** To annotate each amino-acid sequence obtained in transcriptomic and proteomic experiments, we collated the three top hits (together with *E*-values) against the Pfam protein domain database evaluated by HMMER; the top hit against UniProt's UniRef90 database (minimum $E < 0.05$ evaluated by BLASTp and its associated *E*-value; signal peptide prediction results according to SignalP; and the top hit against a database of proteins identified in *P. plagipennis* venom (minimum $E < 0.05$) and its associated per cent identity. The collected annotation data were then reviewed and used to assign proteins into groups based on homology or inferred function according to criteria in Supplementary Table 1. For novel proteins detected in venom by LC-MS/MS, we grouped proteins into families based on BLASTp homology ($E < 0.001$).

**Mass spectrometry.** For LC-MS/MS, 5–50 μg ($A_{280}$ equivalent) of each venom sample or gland extract was reduced with triethylphosphine and alkylated with 2-iodoethanol[41], before digestion with proteomics-grade trypsin (Sigma). Peptide digests were loaded on a Zorbax 300SB-C18 column (Agilent #858750–902) and eluted using a Shimadzu Nexera X2 LC system over a 75 min gradient of 1–40% solvent B (90% acetonitrile and 0.1% formic acid) in solvent A (0.1% formic acid) at a flow rate of 0.2 ml/min. The LC outflow was coupled to a 5600 Triple TOF mass spectrometer (SCIEX) equipped with a Turbo V ion source. MS1 scans were collected between 350 and 2200 *m/z*, and precursor ions in the range *m/z* 350–1500 with charge +2 to +5 and signal >100 counts/s selected for analysis, excluding isotopes within 2 Da. MS/MS scans were acquired with an accumulation time of 250 ms and a cycle time of 4 s. The 'Rolling collision energy' option was selected in the Analyst software, allowing collision energy to be varied dynamically based on *m/z* and *z* of the precursor ion. Up to 20 similar MS/MS spectra were pooled from precursor ions differing by <0.1 Da. The resulting mass spectra in WIFF format were then compared to a library of possible protein sequences generated by extracting all ORFs >90 bp from contigs pooled from all gland compartments of the RNA-Seq experiment, together with a list of common MS contaminants[22] using a Paragon 4.0.0.0 algorithm in ProteinPilot 4.0.8085 software. Minimum criteria used to identify proteins from venom samples were three or more peptides observed with >95% confidence ($p > 95$%), or one or more peptide $p > 95$% plus a secretion signal sequence with *D*-score >0.7 according to SignalP 4.1[39]. False discovery rate (FDR) analysis indicated <1% global FDR for all proteins considered.

For protein quantification from venom gland extracts, we selected proteins and peptides observed with >95% confidence and with global FDR <1% at the protein level. Although it is not possible to reliably distinguish venom proteins from glandular 'housekeeping' proteins using sequence features alone, we nevertheless assigned proteins 'putative housekeeping' or 'possible venom/putative non-housekeeping' status, in order to focus results on likely venom proteins and allow normalisation of quantification data. Proteins were classified as possible venom proteins if they possessed a secretion signal peptide (SignalP *D*-score > 0.4), lacked

a C-terminal ER KDEL or HDEL sequence, lacked transmembrane regions predicted by TMHMM and lacked homology ($E < 0.01$) to insect cuticular proteins and collagen; or if they were >95% identical to proteins detected in venom by mass spectrometry. Other proteins were classified as putative housekeeping proteins. The abundance of each protein was then quantified as the average spectral counts of its three highest-count precursor ions with unique sequence and elution time, consistent with the 'best-flyer' hypothesis[42]. Protein quantifications were then expressed as either the proportion of total non-housekeeping precursor counts (for comparing abundances of different proteins within a single gland compartment extract), or normalised to the sum of total housekeeping protein precursor counts (for comparing abundances of different proteins between gland compartments).

For MALDI-TOF MS, each fraction was diluted in MALDI solvent (70% acetonitrile and 0.1% formic acid) and spotted together with the same volume of α-cyano-4-hydroxycinnamic acid (5 mg/ml in MALDI solvent) and analysed on a SCIEX MALDI-TOF 4700 mass spectrometer in positive reflectron mode. Spectra were collected in over an $m/z$ range of 1000–7000, with final spectra representing the accumulation of 10,000 shots.

**Toxicological assays**. Venom toxicity was evaluated by injecting blowflies (*L. cuprina*) or crickets (*A. domesticus*) with gland extracts or venom using a 27 G needle on a 1 ml syringe driven by a hand microapplicator (Burkard Scientific) that allows injection of precise microlitre quantities. The needle was allowed to penetrate ~1 mm into the insect. For paralysis and death assays in blowflies, 1.7 μl of gland protein extract (of a 30 μl extract of both paired gland compartments of one individual; i.e., 5.6% of the venom in the paired compartments in one individual) was injected into the thorax. Flies were scored for paralysis and death at 15 and 60 min. For escape assays in crickets, third-instar nymphs were injected in the abdomen with 1.7 μl venom, venom dilution or water. Crickets were then immediately placed upright in the centre of an upturned lid of a 100 mm Petri dish, and the time taken to exit the dish recorded.

**Statistics**. All statistical tests were performed in Microsoft Excel using the Real Statistics add-in. Since only a single FPKM value was obtained for each contig (due to the high cost of RNA-Seq experiments on multiple tissues) our statistical analysis of transcript abundances focussed on sets of FPKM values representing either (a) sets of transcripts that encode members of a particular protein family (see section 'The AMG and PMG secrete different proteomes' above; see Supplementary Table 1 for family definitions) or (b) sets of transcripts encoding proteins detected by LC-MS/MS in particular venom samples (see section 'Harrassment and electrostimulation yield distinct venoms' above). Since FPKM values deviated from a normal distribution, we used the non-parametric Kruskal–Wallis test to determine if transcript abundances for different sets of contigs shared the same or different median values (i.e., two-tailed test). We analysed eight samples obtained by harassment and six samples obtained by electrostimulation, as well as nine gland extracts, by LC-MS/MS. Since replicates showed similar patterns of protein identification, protein quantification was performed on one representative sample of each type. For each protein, a robust quantification was achieved by averaging spectral counts over the three most abundant peptides detected from that protein with unique sequence and elution time. Pearson's $r$ was used to test the correlation between FPKM values and protein abundance measured by MS, and the corresponding $t$-values used to test the significance of correlation (two-tailed test). For toxicological assays, five crickets were used to test each venom dilution, with the exception of venom obtained by harassment, where three insects were used due to the lower amount of venom available. The area-proportional Venn diagram shown in Figure 2b was produced using eulerAPE software [43].

**Data availability**. Nucleotide and protein sequences discovered in this project were submitted to GenBank and assigned the accession numbers KY030911–KY031319. Raw sequencing reads and transcriptome assemblies were submitted to NCBI's Sequence Read Archive with the BioProject identifier PRJNA409210. Additional data that support the findings of this study are available from the corresponding authors upon reasonable request.

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

## Acknowledgements

We acknowledge financial support from the Australian Research Council (Grant DP130103813 to G.F.K.), the Australian National Health & Medical Research Council (Principal Research Fellowship APP1044414 to G.F.K.) and the University of Queensland (Postdoctoral Fellowship to A.A.W.). We thank people who donated assassin bugs, Alun Jones for help with mass spectrometry, Nyoman Kurniawan for help with MRI imaging, Greg Baillie and Angelika Christ at the Institute for Molecular Bioscience Sequencing Facility, and Geoff Brown (Department of Agriculture, Fisheries and Forestry, Brisbane, Australia) for supply of blowflies.

## Author contributions

A.A.W., G.F.K., B.G.F., E.A.B.U. and D.J.M. conceived the project and designed experiments. A.A.W. performed and analysed transcriptomic and proteomic experiments, analysed data from MRI and µCT scans, and performed toxicological assays on crickets. M.L.M. stained and imaged tissues by CLSM and differential interference contrast, and the resulting data were analysed by M.L.M. and D.J.M. J.J. performed dissections and additional proteomics experiments. V.H. performed toxicological assays on flies. E.A.B.U. analysed tandem mass spectrometry data for protein quantitation. A.S. performed µCT experiments. All authors contributed to writing the manuscript.

## Additional information

**Competing interests:** The authors declare no competing financial interests.

