## [Peer Review File · Nature Communications]

Reviewers' comments:

Reviewer #1 (Remarks to the Author):

The study by Walker and colleagues shows that the assassin bug has two different venom glands for two different venoms - one used for predation and one for defense. I enjoyed reading the manuscript and found it very well-written and clear despite the complexity of some of the issues (e.g. the morphology and function of the venom gland). The methods and analyses are generally appropriate. The subject is, in my opinion, of high interest to the general scientific audience - relevant to researchers in fields as diverse as ecology, biochemistry, entomology and evolution, and thus appropriate for "Nature communications".

Below are some suggestions and remarks. I support the publication of the manuscript, providing the authors can address a few relatively minor issues (or, rather, important issues that should be relatively easy to address).

My main "general" remark is that there is a certain lack of replication in some figures that makes some of the results less robust than they should be. For example, fig 2A shows the amount of RNA per mg tissue, which is used to suggest that the PMG and AMG are tissues that secrete protein. However, even the large differences in poly-A mRNA between the PMG, AMG and AG could be due to chance. This figure is important and thus needs to have sufficient replication to be trustworthy - namely, the authors should perform the experiment with several individuals and present the mean and standard deviation. Importantly, for such quantification to really be meaningful, the extraction of mRNA needs to be done from similar amounts of tissue (wet or dry weight), otherwise the difference between the tissues could be due to the small amount of starting tissue for the AG (extraction yield often depends on the amount of starting material).

Similarly, it is unclear how many replicates were used to produce the RNA-seq and proteomics results (with the exception of figure 5 and supplementary figure 2). Are these a single analysis of pooled venom/tissue from multiple specimens? Are they from a single specimen? Are they replicated data, and if so, how were the robust transcripts or proteins selected? Given the way the RNA-seq and proteomic data support each other, the results are quite convincing as-is - nevertheless the authors suggest some data on "within-sample comparisons" (line 435) - it would be nice to elaborate on that a bit more, especially in light of the large variability between individuals in the venom produced when harassed (Sup fig 2). Ideally, it would be nice to show how different the two venoms are biochemically (e.g. with MS data) across multiple individuals, which would nicely complement the functional data shown in figure 6A.

Specific remarks:

Abstract - line 31 - not sure the diverse roles the venoms play are the reason they are poorly characterized. I would re-phrase this sentence.

Line 93 - not sure what "anatomical substrates" means

Line 118 - I could see the "piston" in the image - but the suggestion that the piston drives the venom expulsion seems to be only a suggestion. Are there are other data (or literature) to support thus "syringe-like" venom expulsion system? In general, I think it would be nice to elaborate a bit in the manuscript on how the venom apparatus works (e.g. does the piston eject the venom in a syringe-like manner? What drives the venom into the "cone" where the piston is found?)

Line 130 - these are not "secretomes" but rather lists of putatively secreted proteins. The method used to infer these lists is nice and robust but I suggest the authors are a bit more cautious here - maybe use something like "putative secretomes" with a brief sentence explaining what they mean.

Line 134 - "TO the appropriate..."

Lin 235 - is the m/z 3976 (as stated in the text) or 3796 (as shown in the figure)?

Figure 1:

- 1) It would be nice to have an image of the entire animal, showing the location of the venom apparatus
- 2) In figure 1C, it is not clear where the venom ducts are connected to the glands. Do they originate from the Hilus, from the AMG or from the PMG?
- 3) The insets on panels E-f, which I assume are supposed to show the orientation of the 2D sections, are very unclear. I think there is a square showing the location of the specific section but I am not sure... these should be enlarged a bit and clarified.
- 4) The AGD (shown in panel H) is not defined in the figure legend. What is it?
- 5) Panel I - define CLSM (confocal?)
- 6) Panel J is really a beautiful image, but is difficult to interpret when so small. Can a larger version of this image be a supplementary file? Also, the authors suggest that the green staining is nerve cells, but on the right hand side of the image the green staining seems much less "nerve-like". There are large 'sac-like' cells with blue nuclei. How specific is this antibody to nerve cells?

Figure 2: I found that having the mRNA/tissue and the fraction of putative secreted proteins side-by-side a bit misleading - made me think that the height of the bars was comparable. I think this should be with two separate graphs.

Figure 3: I don't think the authors refer to figure 3A in the text

Reviewer #2 (Remarks to the Author):

Review Nature Communications

MS: The assassin bug *Pristhesancus plagipennis* produces distinct predatory and defensive venoms in separate gland lumens

Reviewer: Eduardo G P Fox

I am disclosing of my reviewer ID as I think it is the fair procedure in a single-blind reviewing process, plus it allows for more precise clarification of points and expertise. Overall the paper is well-written and straightforward in its claims. The described system of envenomation is interesting, and the manuscript surely adds to the current arthropod toxinology body of literature.

I leave to the discretion of editors what is the best decision based on my report, but I suggest considerable revision of parts. My main concern is ensuring the reported results can be assessed and reproduced by colleagues.

In any case I shall list suggestions that could further improve the manuscript, along with some key points that ought to be further clarified towards some publication-ready version.

MINOR NARRATIVE SUGGESTIONS.

The writing of the paper is generally clear and easy to follow, but I'd suggest avoiding some specific subjective and judgemental passages, plus minor grammar revision throughout.

Examples as below:

line 32. strike out "exquisite"

line 34. strike out "several"

line 39. "consist"+ent

line 48. "mate" probably 'mating'

line 54. "interaction"+s

line 85. "what +is+ the biological role"(...)strike out "is" at end

line 91. "functionally" probably 'funtional'

line 92. and throughout the text -- consider employing Oxford comma which avoids dual interpretation of noun lists; very useful in technical text.

line 93. "imaged" - perhaps 'images'? Not sure here

line 107. strike out "highly"

line 108. better ":" after "piston"

line 134. "compartment"+ to

line 137. why "copious"? better add some proportions reference for the context or just strike that out

line 161&162. "highly" expressed

line 174. strike out "very", perhaps give a more relative term instead of "abundant" (e.g. some rank?)

line 177. strike out "strongly", perhaps 'demonstrate' instead of "indicate"

line 182. "poorly represented" perhaps more precise as 'not abundant'?

line 326. strike our "greatly"

line 368. perhaps 'hot gun glue' sounds more familiar to "hot-melt adhesive"

These are mere examples which are easy to spot. I am no native speaker, thus the manuscript would likely benefit from a proofread by some critical, skilful external colleague of the authors.

MINOR TECHNICAL SUGGESTIONS

Throughout the narrative specific passages which were not clear to me, regarding either

specific terminology, method details enabling reproducibility, or underlying rationale. I present a short list of such passages to be revisited below.

line 33. Although "two" different kinds of secretions was the observed pattern in the present investigation, I'd expect that these insects may regulate their venom secretion according to context, amounting from more than two possible ends in nature. I'd recommend realising such rationale throughout, but here more in the lines of 'adjust venom use in a context-dependent manner' instead of "inject two different kinds of venom".

line 51. The same should be extended to "dual effect of some toxin pharmacologies". The definition of a toxin is vague: chemicals produced by an organism that can harm another organism. In the end of the day toxins are merely chemicals, and any chemical can be used in countless different ways (e.g. solenopsins as a paralytic insecticide, antibiotic, repellent, chemotherapeutic, etc). A polarized interpretation of reality should always be questioned. Therefore I'd suggest incorporating such rationale herein as 'multifasceted effects these chemicals'.

line 53. The use of "optimum" may sound vague or taken too literal by some readers: there is actually no established "optimum" venom and natural selection is always limited to the available choices. Maybe use 'optimised' instead and expand on this throughout.

line 61. The use of trivial names in scientific literature often leads to confusion, thus why clades are preferred. The name "assassin bugs" is often used for different insects and may not be readily recognisable to many readers from other cultures and backgrounds. I suggest using 'Reduviidae (Insecta: Hemiptera)' on this first instance, particularly in this context of phylogenetics. Similar care should be considered elsewhere where trivial name and other taxonomic levels (reduviids; triatomines) seem to have been used as synonyms.

line 61, again. The phylogeny contextualisation could be improved. Firstly a given reference does not discuss nor mentions the age of the clade Reduviidae, thus perhaps cited instead Xie Q, Tian Y, Zheng L, Bu W. 2008. 18S rRNA hyper-elongation and the phylogeny of Euhemiptera

(Insecta: Hemiptera). Mol. Phylogenet. Evol. 47:463–71. I am no specialist for this group. Secondly the use of the term "diverged" implies that the clade somehow separated from another group mentioned, which may not be the best description of a tree showing the mentioned clades. Perhaps replace this phrase with a more specific localisation of the 'Reduviidae clade within Heteroptera, where it surged ca. XYZ million years ago (insert specific reference(s) herein).

My general suggestion: if discussing the evolution of piercing mouthparts is the main goal, perhaps focus on describing diversity/evolution within "Cimicomorpha" and feeding habits using as specific clade names with trivial names in parentheses where feasible.

line 145., and throughout. I am confused with the use of the term "protein family" across the manuscript as it seems to be context-dependent. I suggest mentioning the specific reference database where one can easily locate (e.g. from line 146) the "pore-forming trialysin/reduysin toxin family", or (e.g. from line 147) "venom family 1" to check how many proteins belong in there, plus other details such as biological source. In case "protein families" is being used to refer to just any assembly of annotated sequences discussed, I recommend adjusting each instance to official PFAM entries, KOG terms, or give names as general annotation terms, according to case.

Line 307. I would be interesting if the authors provided uncurated raw reads or sequencing

files prior to the assembly, as supplementary files.

line 349. I strongly recommend depositing voucher specimens from the same colony using in this study in an official entomological collection, as any day tomorrow taxonomic changes/questions could emerge and the original identity of your samples could be reestablished. Cryptic species, for instance, are a growing issue in entomology that complicates species confirmation in published studies.

line 350-352. Different collection methods yielded different venom secretions; this is the main conclusion of this study. Did the authors account for variation of obtained venom profiles within each collection method? How about between different individuals? No mention is given as to how many different individuals were tested, or how many times, and whether their age or even sex were taken into account. Were those factors controlled? Bioassays ought be made under controlled conditions as to provide marginally reproducible results by restricting chance. The collection methods as described imply that conditions were not controlled. It is fundamental that some range of stability of the venom profiles within different methods be mentioned and described, so that readers evaluate the "noise baseline" for the claimed phenomenon and what to expect when directly replicating the assays. What are the actual chances that a significant part of the described differences derive of, for instance, sexual-maturity variations across different individuals and experimental conditions? Perhaps a first venom inoculation by a certain sexually mature male was much more toxic than from young virgin females, and that only when starved; in such case an immediate replication by a reader would fail possibly affecting the conclusions.

line 352. The supplementary videos show the use of tweezers in provoking the bugs which do not seem "featherlight".

line 243-263. Microinjections into living insects is not trivial, and different methods will greatly impact the results. Please explain in details what kind of needles were used in injecting insects, how far were they introduced, and how were small injection volumes controlled, particularly in the nanoliter scale. Irreproducible microinjection methodology is a major issue in bioassays descriptions with insects, e.g. widespread use of nano-injection systems such as FemtoJet which do not really control injected volumes.

line 429-431. I do not think the criteria for defining the tentative contigs as housekeeping were clear enough to be reproduced. "Proteins were assigned as 'housekeeping' or 'non-housekeeping' depending on the presence of a predicted signal peptide, predicted transmembrane regions, and ER [please also define "ER" here] retention signals." This is a point I'd prefer that the authors make clearer in their manuscript, as it is a recurrent issue accross OMICS analyses. Objectively the term housekeeping gene can be used to those genes which are expressed at a fairly constant rate accross different cell types, meaning the main function of their phenotype is regulating homeostatis. Therefore objectively assigning 'housekeeping genes' within any OMICS analysis primarily depends on as many tissue/cell-type/organ-specific transcriptomes as possible, usually not available. How does one then differentiate a housekeeping gene from toxin-related genes thus depends on assumptions that need to be clarified. Specially when minding toxin genes are typically recent duplications of functionally unrelated proteins (e.g. vitellogenin as *Api m 12* in bees and OBPs as *Sol i 2* and *Sol i 4* in fire ants). Some authors deposited microarray comparisons for a few model organisms, while others resorted to counteracting strategies, such as subtractive cDNA libraries from whole bodies or from whole bodies by minus venom glands.

The present study cannot rely on comparative expression from other tissues and employed no specific strategy for housekeeping genes. Please thus define more specifically which criteria were used in filtering out housekeeping sequences, and why, based on relevant references. Would the employed method correctly identify, e.g. Apis m 12 among vitellogenin ESTs? This is a fundamental point in venom OMICs studies which is frequently overlooked. Finally I should remark that likely the obtained contigs do not represent complete transcripts, thus an absence of e.g. a signal peptide does not necessarily mean there is none in the actual transcript.

line 410. I miss a description of details on how the contigs were annotated, including search database and algorithm settings, in a way which would enable reproducing the steps later or in related projects.

line 455 and throughout. No software is declared for the statistical analysis; please clarify. It is known that different programs might affect analyses results. Moreover, I strongly suggest using open sourced software such as R which directly enable including analytical and plot scripts as a supplementary text file. Such relatively simple addition will greatly improve transparency and add value to published data which can be readily reevaluated. For instance, one could easily check whether conclusions would be affected if data had been e.g. square-root-transformed enabling the use of parametric tests, which might have been the preferred route of some other authors. Based on the appearance of the plots perhaps R was already employed, meaning the scripts might be already available.

Finally I am thankful for the opportunity to contribute on an interesting research report. I hope my suggestions and questions contribute to some final version of the present manuscript. I remain open to any communication with the authors on the topic, as relates with my line of research.

My best regards,

Eduardo

Reviewer #3 (Remarks to the Author):

This communication follows previous work of these authors on tissue specialized venom production in various toxic animals. The main point, besides immense information on new proteinaceous compounds, is outlined in the title "The assassin bug *Pristhesancus plagipennis* produces distinct predatory and defensive venoms in separate gland lumens". This observation extends previous work indicating that selective pressures drove the speciation of cone snail venomous duct tissue into two functionally distinct regions (Dutertre et al 2014 Nature Commun. 5; ref 7). In the present paper, the posterior venomous gland of the bug produces a toxin mixture used mostly for hunting, whereas the anterior venomous gland produces mainly a protein mix for defense (paralytic and pain inducing compounds). The data, based on protein chemistry, transcriptomic, anatomic, physiologic and toxicologic analyses, is highly convincing and was carried out very carefully and thoughtfully. It should be mentioned, however, that the findings of cell specialized secretion of different components in endocrine glands of animals is not new. For example,

distinct venomous secretory cells beside nematocysts that produce toxins for predation (usually in tentacles) and others for defense (at the body wall) have been shown in sea anemones (Moran et al 2013 *Mar Biotechnol* 15, 329). Scorpions have also been described to inject venom of various composition upon stinging as a warning defensive signal versus toxins used for predation (Inceoglu et al 2003 *Proc Natl Acad Sci USA* 100, 922); Various toxins that appear in organs other than the venom gland were shown by various researchers indicating specialization of secretory cells and localization at various regions of cone snails (Bia et al, 2009 *J Proteomics* 72, 210; Biggs et al 2008 *Toxicon* 52, 101; Safavi-Hemami et al 2011 *J Biol Chem* 286, 22546). Thus, the main contribution of the present paper is the 'state of art' experimental approach showing specialization of two distinct venom glands, which secrete upon different stimuli. Defining of the nerve circuit that controls the muscles of each gland might be a great achievement, and should be considered in future work.

Besides the data, the paper is well written and organized including the graphic arts, supporting data and accompanying videos.

Revisions undertaken on manuscript NCOMMS-17-18024 and response to Reviewers

October 2017

We thank the Reviewers for their insightful comments and have undertaken substantial revisions to this manuscript in response to their suggestions. These revisions include both editing of the manuscript (e.g. to clarify methodological details) as well as new experimental data. The additional experimental data provided in this revision are as follows:

- Proteomic analysis of gland compartment extracts is now included for nine samples from three individuals instead of three samples from one individual (Supplementary Data S2, Sheets A–M).
- Proteomic analysis of venom samples obtained by harassment is now included for eight samples from different individuals instead of one sample from one individual (Supplementary Data S3, Sheets A, C–J).
- Proteomic analysis of venom samples obtained by electrostimulation is now presented for six samples from different individuals instead of one sample from one individual (Supplementary Data S3, Sheets B, K–P).
- MALDI spectra of six venom samples from different individuals obtained by electrostimulation are now presented, in order to illustrate the clear difference in their spectra compared to the spectra of eight venom samples from different individuals obtained by harassment (Extended Data Figure E3).

Additional data has also been provided to improve data transparency:

- Numerical data used to compare transcript abundance using the Kruskal-Wallis tests are now appended as an additional worksheet in Supplementary Data S1 (Sheet B).
- Numerical data used to compare transcript abundance and protein abundance in each gland compartment (presented in Figure 4) is now appended in Supplementary Data S4 (Sheets A–D).
- Unambiguous criteria for assigning proteins into groups are now given (Supplementary Table S1).

A detailed response to each issue raised is given below (in blue text).

Reviewer #1 (Remarks to the Author):

The study by Walker and colleagues shows that the assassin bug has two different venom glands for two different venoms - one used for predation and one for defense. I enjoyed reading the manuscript and found it very well-written and clear despite the complexity of some of the issues (e.g. the morphology and function of the venom gland). The methods and analyses are generally appropriate. The subject is, in my opinion, of high interest to the general scientific audience - relevant to researchers in fields as diverse as ecology, biochemistry, entomology and evolution, and thus appropriate for "Nature communications".

We thank the reviewer for these kind comments.

Below are some suggestions and remarks. I support the publication of the manuscript, providing the authors can address a few relatively minor issues (or, rather, important issues that should be relatively easy to address).

My main "general" remark is that there is a certain lack of replication in some figures that makes some of the results less robust than they should be. For example, fig 2A shows the amount of RNA per mg tissue, which is used to suggest that the PMG and AMG are tissues that secrete protein. However, even the large differences in poly-A mRNA between the PMG, AMG and AG could be due to chance. This figure is important and thus needs to have sufficient replication to be trustworthy - namely, the authors should perform the experiment with several individuals and present the mean and standard deviation. Importantly, for such quantification to really be meaningful, the extraction of mRNA needs to be done from similar amounts of tissue (wet or dry weight), otherwise the difference between the tissues could be due to the small amount of starting tissue for the AG (extraction yield often depends on the amount of starting material).

To address the issue of replication generally, we have included extensive additional proteomic data as mentioned above. However, our RNA-Seq data is based on three sequencing experiments—a single experiment from each of three gland compartments—as replicate experiments over each of the three tissues was prohibitively expensive. For this reason we employed several strategies to increase the robustness of our results, including (a) combination of both transcriptomic and proteomic techniques, and (b) sequencing RNA derived from tissue from a pool of individuals. In the case of this specific experiment, the mRNA quantitation was performed during the course of library preparation and hence represents a single measurement of tissue from a pool of individuals. Additionally, we have provided these data in units of ng RNA *per mg starting tissue*, in order that the weight of starting tissue does not affect the results. To clarify these points we altered the first sentence of Results section 2.2 (revised manuscript lines 147–150) to read:

“To examine the secretory output of each gland compartment we utilised previously assembled transcriptomes of the PMG, AMG and AG²². Each of these transcriptomes represents a single sequencing experiment conducted on mRNA from a specific gland compartment, using mRNA of glandular tissue pooled from four individuals.”

The *RNA-Seq and transcriptomics* subsection of the Methods section (revised manuscript lines 459–462) was revised to read:

“RNA-Seq and transcriptome assembly was performed as previously described²². Transcriptomes of the AMG, PMG, and AG were generated from four individuals (two adult females and two last-instar nymphs) by pooling gland compartments between individuals.”

We agree with Reviewer 1 that it would be ideal to present mRNA quantitation data for the three gland compartments replicated with standard deviations. However, we have so far been unable to collect this data due to lack of experimental animals. We would need eight more *Pristhesancus plagipennis* to replicate this data at $n = 3$. Currently we do not have any and are unlikely to be able to collect eight adults until late in the Australian summer. To negate this problem, we have recently attempted to replace this data with mRNA quantitation data from each compartment of three single individuals. However, these experiments were unsuccessful due to the small amount of starting material and possibly other factors. Therefore (and due to the below comments about presenting both measures of secretory output on one graph; see below) we have removed mRNA quantitation data from Figure 2. We have also altered the text to add the appropriate caveats to these results (e.g. changing ‘indicate’ to ‘suggest’) on line 160 of the revised manuscript. Lines 160–164 were altered to read:

“These data suggest that the AMG and PMG are the main secretory tissues of the venom glands, which is consistent with the high yield of polyA(+) mRNA obtained from the AMG and PMG (356 and 226 ng per mg tissue respectively) but not AG (13 ng per mg tissue) obtained during library preparation²².”

Overall we think this is an acceptable solution in the context of the other data presented in the manuscript.

Similarly, it is unclear how many replicates were used to produce the RNA-seq and proteomics results (with the exception of figure 5 and supplementary figure 2). Are these a single analysis of pooled venom/tissue from multiple specimens? Are they from a single specimen? Are they replicated data, and if so, how were the robust transcripts or proteins selected?

For transcriptomic data, clarifications added to the manuscript concerning replicates and number of animals is given above. For proteomics data, we have now provided additional experimental data to resolve this issue (see above). We have also edited the text in several places to more clearly indicate the numbers of animals used and in what manner in the following passages:

Lines 193–195:

“The PMG, AMG, and AG were extracted from three adult bugs and the protein content of the nine extracts analysed individually using LC-MS/MS.”

Lines 210–213:

“Results were similar in the remaining two individuals analysed: proteases and redulysins were more confidently identified than hemolysin-like proteins in the PMG, whereas the reverse was true in the AMG (Supplementary Data S2, Sheets E–M).”

Lines 244–252:

“We further examined the contribution of between-individual variability to this result by performing separate Paragon searches for each of eight samples of venom obtained by harassment and six samples obtained by electrostimulation. All eight venom samples obtained by harassment showed similar profiles in which hemolysin-like proteins were identified with higher confidence than proteases or redulysins (Supplementary Data S3, Sheets C–J). The reverse was true for six samples obtained by electrostimulation (Supplementary Data S3, Sheets K–P). Thus, samples collected from *P. plagiipennis* differ markedly depending on the method by which they are collected.”

Lines 275–278:

“MALDI-TOF mass spectra of six individual venom samples obtained by electrostimulation were similar (Extended Data Figure E3A), whereas eight individual samples obtained by harassment were more variable, despite all experimental animals being siblings (Extended Data Figure E3B).”

Given the way the RNA-seq and proteomic data support each other, the results are quite convincing as-is - ...

As Reviewer #1 suggests, another strategy we employed to improve the robustness of our findings was to examine the secretory output of each gland using both transcriptomic and proteomic techniques. To clarify our experimental rationale to the reader, we have altered the third paragraph of Section 2.2 (revised manuscript lines 191–193) to read:

“We investigated if differences observed in transcript abundance between compartments in RNA-Seq experiments corresponded to differences in the abundances

of proteins accumulated in each gland lumen.”

... nevertheless the authors suggest some data on “within-sample comparisons” (line 435) - it would be nice to elaborate on that a bit more, especially in light of the large variability between individuals in the venom produced when harassed (Sup fig 2).

The mention of “within-sample comparisons” (original submission line 435) was not meant to refer to experiments examining variation between individuals, but comparing different proteins identified within a single venom sample. We have altered this passage to clarify its meaning (revised manuscript lines 515–520):

“Protein quantifications were then expressed as either the proportion of total non-housekeeping precursor counts (for comparing abundances of different proteins within a single gland compartment extract), or normalised to the sum of total housekeeping protein precursor counts (for comparing abundances of different proteins between gland compartments).”

The individual sample data provided in Supplementary Data S3 (Sheets C–P) and Extended Data Figure E3 show that while there is much variation at the peptide level, the larger proteins present in venom obtained by harassment show less inter-individual variability, with hemolysin-like proteins and ‘family 17’ proteins always well-represented in venom from the AMG.

Ideally, it would be nice to show how different the two venoms are biochemically (e.g. with MS data) across multiple individuals, which would nicely complement the functional data shown in figure 6A.

As described above, we appended additional MS experimental results to address this issue.

Specific remarks:

Abstract - line 31 - not sure the diverse roles the venoms play are the reason they are poorly characterized. I would re-phrase this sentence.

We have rephrased this sentence (revised manuscript lines 30–31) to read:

“The assassin bug venom system plays diverse roles in prey capture, defense, and extra-oral digestion, but it is poorly characterised, partly due to its anatomical complexity.”

Line 93 - not sure what “anatomical substrates” means

We altered the last sentence of the Introduction (revised manuscript lines 102–105) to replace this term with a clearer alternative:

“However, in contrast to these two latter groups, assassin bugs possess a set of anatomical structures clearly suited to the delivery of multiple complex venoms, as we demonstrate in this study.”

Line 118 - I could see the “piston” in the image - but the suggestion that the piston drives the venom expulsion seems to be only a suggestion. Are there are other data (or literature) to support thus “syringe-like” venom expulsion system? In general, I think it would be nice to elaborate a bit in the manuscript on how the venom apparatus works (e.g. does the piston eject the venom in a syringe-like manner? What drives the venom into the “cone” where the piston is found?)

The syringe-like or piston-like workings of the hemipteran venom or salivary pump are not known in detail but have been proposed based on morphological grounds. In the case of *P. plagiipennis*, the contraction of muscle fibres surrounding the venom glands most likely is responsible for driving venom into the venom pump. To make these points clearer, we have altered the Results Section 2.1 and inserted a reference describing this structure in a plant feeding hemipteran (now reference 25): Raine J, Forbes AR. 2012. The salivary syringe of the leafhopper *Macrostelus fascifrons* (Homoptera: Cicadellidae) and the occurrence of Mycoplasma-like organisms in its ducts. *Canad. Entomol.* 103:110–116. Lines 131–138 of the revised manuscript now read:

“The VP is a hollow, conical structure with a large muscle bundle (MB) connected to the centre of the concavity (Figure 1, L–N), an arrangement noted in other hemipterans to resemble a piston^{25,26}: muscle contraction will fill the pump cavity, while relaxation of the bundle will cause expulsion of the venom through the venom channel (VC) that extends through the ventral part of the proboscis and ultimately into the stylet tips. We did not observe a ‘complex valve’ as exists at the junction of the VP and VC in giant water bugs (Belostomatidae)²⁷, suggesting its absence in reduviids.”

With regards to further elaboration on how the overall venom system functions, the first paragraph of the Discussion (lines 330–336) was altered to read:

“The details of how each component of the venom system functions during envenomation remain to be elucidated, but most likely the glandular valves, muscle networks and venom pump are activated by CNS input in a co-ordinated manner to inject the appropriate venom. These results also explain why harvesting venom using electrostimulation is not always successful or instantaneous, and why it does not expel the contents of all venom gland compartments, since electrically induced muscle

contraction might either restrict or facilitate venom flow at multiple points.”

Line 130 - these are not “secretomes” but rather lists of putatively secreted proteins. The method used to infer these lists is nice and robust but I suggest the authors are a bit more cautious here - maybe use something like “putative secretomes” with a brief sentence explaining what they mean.

We have altered the first paragraph of Section 2.2 (lines 150–154) to address these issues:

“To produce putative *secretomes* (i.e., lists of the putative secreted proteins in each compartment), we filtered transcriptomes for contigs encoding open reading frames with a high-confidence secretion signal peptide, putative coding region >150 bp, and stop codon, but without ER retention signal or predicted transmembrane regions (see Methods).”

Line 134 - “TO the appropriate...”

TO has been replaced with ‘to’ in this instance.

Lin 235 - is the m/z 3976 (as stated in the text) or 3796 (as shown in the figure)?

The text has been altered to read the correct value, 3796 Da (revised manuscript line 283).

Figure 1:

1) It would be nice to have an image of the entire animal, showing the location of the venom apparatus

We agree and for this purpose we have labelled the external surface of the insect in these MRI scans (shown in the first two panels). The entire insect is shown except for the posterior tip of the abdomen, which was outside the region of the scan. To make this clearer to the reader we altered the legend of Figure 1A and 1B (revised manuscript lines 699–700):

“(A) Sagittal view and (B) dorsal view, showing the position of glands in relation to the external surface of the insect (light green).”

2) In figure 1C, it is not clear where the venom ducts are connected to the glands. Do they originate from the Hilus, from the AMG or from the PMG?

The venom ducts originate from the hilus. An enlargement is shown in 1H. To make this clearer we altered the figure legend to read (revised manuscript lines 702–703):

“(C) Dorsal view of main gland complex. VD, venom ducts connecting the hilus to venom pump (pink); H, hilus (green).”

3) The insets on panels E-f, which I assume are supposed to show the orientation of the 2D sections, are very unclear. I think there is a square showing the location of the specific section but I am not sure... these should be enlarged a bit and clarified.

We agree, and have enlarged to three times previous size in each case. We also updated the legend to read (revised manuscript lines 705–707):

“(E–G) 2D slices from MRI, with compartments in upper panels coloured dark blue (AMG), red (PMG) and green (hilus). Insets show location of each slice.”

4) The AGD (shown in panel H) is not defined in the figure legend. What is it?

The AGD is now defined in the legend (line 704 of revised manuscript) as the accessory gland duct.

5) Panel I - define CLSM (confocal?)

The legend of panel I has been altered to replace ‘CLSM’ with ‘Confocal laser scanning microscopy’ (line 711, revised manuscript).

6) Panel J is really a beautiful image, but is difficult to interpret when so small. Can a larger version of this image be a supplementary file?

We have now provided the unprocessed image as Extended Data Figure E1. We also added some insets to Figure 1J, showing enlargements of the nerve processes marked by black arrows, to help readers interpret this image.

Also, the authors suggest that the green staining is nerve cells, but on the right hand side of the image the green staining seems much less “nerve-like”. There are large ‘sac-like’ cells with blue nuclei. How specific is this antibody to nerve cells?

The antibody used is specific for acetylated tubulin, but acetylated tubulin is present in cells other than neurons. The neuronal processes in Figure 1J are identifiable because of their rich staining for acetylated tubulin but also their characteristic appearance. To clarify this, we updated Section 2.1 (lines 125–130 of revised manuscript) to read:

“An antibody against acetylated tubulin strongly stained structures with the characteristic appearance of nerve processes. These processes, supported by trachea

attached to the venom duct, extensively innervate muscle fibres close to the hilus and on the surface of the glands (Figure 1J, Extended Data Figure E1) revealing a potential for controlling venom outflow as summarised in Figure 1K.”

The figure legend (revised manuscript lines 715–720) was altered to read:

“(J) CLSM and differential interference contrast (DIC) images showing hilus and surrounding structures. Left, DIC image. Right, antibody against acetylated tubulin (green) highlights nerves (N) and neuromuscular junctions (black arrows) as well as fibrous-appearing staining near the start of VD (white arrows) and gland cells. Red: actin staining of muscle fibres. Blue: nuclear stain. Insets: enlargements of neuromuscular junctions marked by black arrows.”

Figure 2: I found that having the mRNA/tissue and the fraction of putative secreted proteins side-by-side a bit misleading - made me think that the height of the bars was comparable. I think this should be with two separate graphs.

As we have removed the mRNA quantitation data from Figure 2A (see comment above), this issue has been negated.

Figure 3: I don't think the authors refer to figure 3A in the text

Thanks for noting this omission. A reference has been added to Figure 3A on line 172 of the revised manuscript.

Reviewer #2 (Remarks to the Author):

Review Nature Communications

MS: The assassin bug *Pristhesancus plagipennis* produces distinct predatory and defensive venoms in separate gland lumens

Reviewer: Eduardo G P Fox

I am disclosing of my reviewer ID as I think it is the fair procedure in a single-blind reviewing process, plus it allows for more precise clarification of points and expertise.

Overall the paper is well-written and straightforward in its claims. The described system of envenomation is interesting, and the manuscript surely adds to the current arthropod toxinology body of literature.

Thank you for your kind comments.

I leave to the discretion of editors what is the best decision based on my report, but I suggest considerable revision of parts. My main concern is ensuring the reported results can be assessed and reproduced by colleagues.

In any case I shall list suggestions that could further improve the manuscript, along with some key points that ought to be further clarified towards some publication-ready version.

MINOR NARRATIVE SUGGESTIONS.

The writing of the paper is generally clear and easy to follow, but I'd suggest avoiding some specific subjective and judgemental passages, plus minor grammar revision throughout.

Examples as below:

line 32. strike out "exquisite"

The phrase "... a series of exquisite adaptations ..." has been replaced with the phrase "... numerous adaptations ..." (revised manuscript line 32).

line 34. strike out "several"

The word "several" was replaced with "three" (revised manuscript line 34).

line 39. "consist"+ent

The word "consist" was replaced with "consistent" (revised manuscript line 40).

line 48. "mate" probably 'mating'

The word "mate" was replaced with "mating" (revised manuscript line 48).

line 54. "interaction"+s

The word “interaction” was replaced with “interactions” (revised manuscript line 54).

line 85. "what +is+ the biological role"(...)strike out "is" at end

This change appears to us to produce a non-grammatical sentence so we replaced the entire sentence to an alternative wording that we think is better (revised manuscript lines 93–95):

“However, neither the gland compartment that produces this venom nor the biological role of each compartment is known.”

line 91. "functionally" probably 'funtional'

The word “functionally” was replaced with “functional” (revised manuscript line 101).

line 92. and throughout the text -- consider employing Oxford comma which avoids dual interpretation of noun lists; very useful in technical text.

We previously avoided the Oxford comma throughout the manuscript as we were under the impression this is the journal preference, but having checked we can find no journal preference. We have inserted the comma in this instance.

line 93. "imaged" - perhaps 'images'? Not sure here

Does this comment refer to (original submission) line 99, as there is no instance of the word “imaged” on line 93? If so, we think “magnetic resonance imaging” is the appropriate phrase, since we are referring to the technique, and this is the standard name for the technique (revised manuscript line 110).

line 107. strike out "highly"

This word has been removed (revised manuscript lines 117–118).

line 108. better ":" after "piston"

This passage has been re-written in response to this comment and those of Reviewer 1. It now carries a colon after “piston” in accordance with this comment (revised manuscript lines 131–138).

line 134. "compartment"+ to

The word “to” has been inserted where indicated (revised manuscript line 158).

line 137. why "copious"? better add some proportions reference for the context or just strike that out

This section has been rewritten in response to comments of Reviewer 1 and no longer contains the word ‘copious’. The new passage can be found in lines 160–164 of the revised manuscript.

line 161&162. "highly" expressed

The word “strongly” was changed to “highly” in both instances (revised manuscript lines 187–189).

line 174. strike out "very", perhaps give a more relative term instead of "abundant" (e.g. some rank?)

We have re-written this sentence and the following one to more exactly describe these data (lines 206–210):

“In the AMG, hemolysin-like proteins (29%), cystatins (14%), and unknown proteins (39%) were the three most abundant classes. Of all non-housekeeping precursor counts detected from the AMG, 29% originated from a single protein in the unknown class, which is also the product of the highest-abundance transcript in the AMG secretome (secretome ID #28).”

line 177. strike out "strongly", perhaps 'demonstrate' instead of "indicate"

The phrase “strongly indicate” has been replaced with “demonstrate” (revised manuscript lines 213–215).

line 182. "poorly represented" perhaps more precise as 'not abundant'?

We rewrote the opening of Results Section 2.4 to read (revised manuscript lines 219–222):

“Venom secreted by the AMG is not abundant in venom harvested by electrostimulation. For example, for the 34 most highly expressed secretome contigs in the AMG (FPKM 100–21,859), most protein products (24 proteins or 71%) were not detected in venom harvested by electrostimulation²².”

line 326. strike our "greatly"

The word "greatly" was removed (revised manuscript line 380).

line 368. perhaps 'hot gun glue' sounds more familiar to 'hot-melt adhesive'

The phrase "with hot-melt adhesive" was replaced with "with a hot glue gun" (revised manuscript 431–432).

These are mere examples which are easy to spot. I am no native speaker, thus the manuscript would likely benefit from a proofread by some critical, skilful external colleague of the authors.

MINOR TECHNICAL SUGGESTIONS

Throughout the narrative specific passages which were not clear to me, regarding either specific terminology, method details enabling reproducibility, or underlying rationale. I present a short list of such passages to be revisited below.

line 33. Although "two" different kinds of secretions was the observed pattern in the present investigation, I'd expect that these insects may regulate their venom secretion according to context, amounting from more than two possible ends in nature. I'd recommend realising such rationale throughout, but here more in the lines of 'adjust venom use in a context-dependent manner' instead of "inject two different kinds of venom".

We altered this sentence (revised manuscript lines 31–34) so that it now reads:

"We demonstrate that this complexity results from numerous adaptations that enable assassin bugs to modulate the composition of their venom in a context-dependent manner."

line 51. The same should be extended to "dual effect of some toxin pharmacologies". The definition of a toxin is vague: chemicals produced by an organism that can harm another organism. In the end of the day toxins are merely chemicals, and any chemical can be used in countless different ways (e.g. solenopsins as a paralytic insecticide, antibiotic, repellent, chemotherapeutic, etc). A polarized interpretation of reality should always be questioned. Therefore I'd suggest incorporating such rationale herein as 'multifasceted effects these chemicals'.

We have replaced the phrase "... numerous different toxins and the dual effects of some toxin pharmacologies ..." with "... numerous different molecules in venom, and their

multifaceted pharmacological effects ...” (revised manuscript lines 50–51).

line 53. The use of "optimum" may sound vague or taken too literal by some readers: there is actually no established "optimum" venom and natural selection is always limited to the available choices. Maybe use 'optimised' instead and expand on this throughout.

We re-wrote this sentence in a different way that makes the same point but hopefully negates this confusion (revised manuscript lines 53–55):

“Nevertheless, some venoms will be better suited to certain interactions than others depending on their composition, which might result in selection for the ability to vary venom composition.”

line 61. The use of trivial names in scientific literature often leads to confusion, thus why clades are preferred. The name "assassin bugs" is often used for different insects and may not be readily recognisable to many readers from other cultures and backgrounds. I suggest using 'Reduviidae (Insecta: Hemiptera)' on this first instance, particularly in this context of phylogenetics. Similar care should be considered elsewhere where trivial name and other taxonomic levels (reduviids; triatomines) seem to have been used as synonyms.

line 61, again. The phylogeny contextualisation could be improved. Firstly a given reference does not discuss nor mentions the age of the clade Reduviidae, thus perhaps cited instead Xie Q, Tian Y, Zheng L, Bu W. 2008. 18S rRNA hyper-elongation and the phylogeny of Euhemiptera

(Insecta: Hemiptera). *Mol. Phylogenet. Evol.* 47:463–71. I am no specialist for this group. Secondly the use of the term "diverged" implies that the clade somehow separated from another group mentioned, which may not be the best description of a tree showing the mentioned clades. Perhaps replace this phrase with a more specific localisation of the 'Reduviidae clade within Heteroptera, where it surged ca. XYZ million years ago (insert specific reference(s) herein).

My general suggestion: if discussing the evolution of piercing mouthparts is the main goal, perhaps focus on describing diversity/evolution within "Cimicomorpha" and feeding habits using as specific clade names with trivial names in parentheses where feasible.

We agree there were several unclear points in this passage. We have re-written the first part of this paragraph as suggested to both improve the phylogenetic and taxonomic treatment of insect groups, and to focus on the point we are trying to make here, which is understanding the structure of the assassin bug venom gland in its evolutionary context (revised manuscript lines 66–75):

“Assassin bugs are the predatory, venomous insects in family Reduviidae (Insecta: Hemiptera), excluding the kissing bugs (subfamily Triatominae) that feed on vertebrate

blood. Reduviidae is part of suborder Heteroptera, a group that diverged from other hemipteran insects such as cicadas and aphids ~245 mya¹¹, co-incident with a trophic shift from phytophagy to predation¹². The feeding and salivary system of their phytophagous forebears, comprising complex labial glands and piercing-sucking mouthparts, was adapted by heteropterans to form a venom apparatus. The overall arrangement of the glandular structures nevertheless remains similar: hemipteran labial glands are paired structures in the thorax and abdomen, comprising the anterior (AMG) and posterior lobes (PMG) of the main gland plus an accessory gland (AG)¹²."

We have also replaced reference 9 (original submission) with a more appropriate one (revised manuscript reference 11, Misof, B. *et al.* Phylogenomics resolves the timing and pattern of insect evolution. *Science* **346**, 763–767, 2014) and inserted family names for the insects discussed in the rest of this paragraph (Lygaeidae, Asopinae and Belostomatidae; see revised manuscript lines 79 and 83) and in the Discussion (Triatominae and Cimicidae; revised manuscript lines 372–373).

line 145., and throughout. I am confused with the use of the term "protein family" across the manuscript as it seems to be context-dependent. I suggest mentioning the specific reference database where one can easily locate (e.g. from line 146) the "pore-forming trialysin/redulyisin toxin family", or (e.g. from line 147) "venom family 1" to check how many proteins belong in there, plus other details such as biological source. In case "protein families" is being used to refer to just any assembly of annotated sequences discussed, I recommend adjusting each instance to official PFAM entries, KOG terms, or give names as general annotation terms, according to case.

Here and elsewhere, these protein families actually correspond to our manual grouping of proteins into families based on multiple sequence features and annotation sources, including the HMMER algorithm against the Pfam database, BLAST searches against the UniRef90 and GenBank's protein nr databases, BLAST searches against internal datasets (i.e. to group unknown proteins) and other sequence features. This approach was taken because we needed to discuss and analyse not only known proteins, but also unknown groups of proteins, and because different criteria are appropriate to different protein groups. To clarify exactly our definitions and criteria for each group without bogging down the text and affecting its readability, we have supplied unambiguous criteria used to group proteins in this study as Supplementary Table S1. We also inserted an additional subsection 'Sequence annotation' into the Methods, and altered text in three places in the Results section to direct readers to the appropriate definitions. The new subsection in the Methods section (revised manuscript lines 475–485) now reads:

“Sequence annotation

To annotate each amino acid sequence obtained in transcriptomic and proteomic experiments, we collated the three top hits (together with *E*-values) against the Pfam protein domain database evaluated by HMMER; the top hit against UniProt’s UniRef90 database (minimum *E* < 0.05) evaluated by BLASTp and its associated *E*-value; signal peptide prediction results according to SignalP; and the top hit against a database of proteins identified in *P. plagipennis* venom (minimum *E* < 0.05) and its associated percent identity. The collected annotation data was then reviewed and used to assign proteins into groups based on homology or inferred function according to criteria in Supplementary Table S1. For novel proteins detected in venom by LC-MS/MS, we grouped proteins into families based on BLASTp homology (*E* < 0.001).”

The first mention of annotation in regard to transcriptomic data (revised manuscript lines 168–171) now reads:

“To further investigate secretory output of each gland compartment, we annotated each putative secreted sequence using HMMER and BLAST searches, and proteins were classified into groups based on homology or inferred function (see Methods; Supplementary Data S1).”

The first mention of annotation in regard to proteomics of gland extracts (revised manuscript lines 197–198) now reads:

“Each identified protein was also annotated using the HMMER, BLAST, SignalP and TMHMM algorithms (Supplementary Data S2, Sheets A–D; see Methods).”

We altered the description of proteins discovered in venom harvested by harassment so that lines 232-236 read:

“A pooled Paragon search of mass spectra from all eight samples (against open reading frames from all gland compartments) yielded a total of 150 proteins, of which 72 were not previously reported in *P. plagipennis* venom²². Identified protein sequences were annotated by HMMER and BLAST searches (Supplementary Data S3, Sheet A; see Methods).”

Line 307. I would be interesting if the authors provided uncurated raw reads or sequencing files prior to the assembly, as supplementary files.

The raw sequencing reads and assembled transcriptomes used in this study have now been deposited to NCBI’s Sequence Read Archive and are now available for download. We added a sentence indicating this following the protein sequence Accession numbers submitted to GenBank, in the Acknowledgements section (revised manuscript lines 573–575):

“Raw sequencing reads and transcriptome assemblies were submitted to NCBI’s Sequence Read Archive with the BioProject identifier PRJNA409210.”

line 349. I strongly recommend depositing voucher specimens from the same colony using in this study in an official entomological collection, as any day tomorrow taxonomic changes/questions could emerge and the original identity of your samples could be reestablished. Cryptic species, for instance, are a growing issue in entomology that complicates species confirmation in published studies.

We have now deposited two voucher specimens (male and female) to the Queensland Museum. We have also provided the reference (revised manuscript reference 34: Malipatil M. 1986. *Australian Journal of Zoology*. 34:601–610) containing the key used to identify wild-caught insects used in this study. This information was inserted into the first paragraph of the Methods section: “Insects, dissections and venom collection”. Lines 406–411 of the revised manuscript now read:

“Assassin bugs (*Pristhesancus plagipennis*) were collected in Brisbane, Australia, and fed on crickets (*A. domesticus*; Pisces Enterprises, Kenmore, QLD, Australia). Both wild-caught and captive-bred animals were used in this study. Species-level identification was performed according to a published key³⁴ and two voucher specimens were deposited in the Queensland Museum Entomology Collection with reference numbers T239616 (male) and T239617 (female).”

line 350-352. Different collection methods yielded different venom secretions; this is the main conclusion of this study. Did the authors account for variation of obtained venom profiles within each collection method? How about between different individuals? No mention is given as to how many different individuals were tested, or how many times, and whether their age or even sex were taken into account. Were those factors controlled? Bioassays ought be made under controlled conditions as to provide marginally reproducible results by restricting chance. The collection methods as described imply that conditions were not controlled. It is fundamental that some range of stability of the venom profiles within different methods be mentioned and described, so that readers evaluate the "noise baseline" for the claimed phenomenon and what to expect when directly replicating the assays. What are the actual chances that a significant part of the described differences derive of, for instance, sexual-maturity variations across different individuals and experimental conditions? Perhaps a first venom inoculation by a certain sexually mature male was much more toxic than from young virgin females, and that only when starved; in such case an immediate replication by a reader would fail possibly affecting the conclusions.

To address this question we have provided replicates of venom samples from different individuals (eight by harassment, six by electrostimulation) as described above. As can be seen from these data (Supplementary Data S3, sheets C–P), venom composition is strongly affected by collection method, and this is a highly robust result. Lines 244–252 of the revised manuscript have been altered to indicate this clearly:

“We further examined the contribution of between-individual variability to this result by performing separate Paragon searches for each of eight samples of venom obtained by harassment and six samples obtained by electrostimulation. All eight venom samples obtained by harassment showed similar profiles in which hemolysin-like proteins were identified with higher confidence than proteases or redulysins (Supplementary Data S3, Sheets C–J). The reverse was true for all six samples obtained by electrostimulation (Supplementary Data S3, Sheets K–P). Thus, samples collected from *P. plagipennis* differ markedly depending on the method by which they are collected.”

We have also added further details to the Methods section to describe the controlled conditions under which venom was extracted. Revised manuscript lines 412–415 now read:

“Venom was harvested from adult bugs more than one week after their final moult, and 4–6 days after feeding. Venom harvest was by electrostimulation²² or by gently harassing awake but restrained bugs by touching legs, antennae and abdomen with a pair of tweezers to simulate attack by a small animal. Venom was immediately transferred to a tube on ice and stored at –20°C until analysis.”

Due to the scarcity of the animals and the large number required for this study, we did not analyse differences in venom composition depending on (a) mating status of the animal, or (b) gender. Mating status is unknown for bugs collected as adults, except in the case of a female that subsequently lays eggs. Sex can frequently be determined by the size of an individual and the shape of its abdomen, but the sex of some individuals is ambiguous without dissection. For this reason, some venom samples were obtained without recording a definite sex of the bug from which it came. In general, no obvious differences were observed between males and females. To make these limitations in our study clear to the reader, we inserted two sentences into the end of the Discussion (revised manuscript lines 390–393):

“A limitation to our study is that we have not analysed differences in venom composition between males and females, or between the different immature and adult stages, or between virgin and mated individuals. Additional studies may identify further venom peptides and proteins, and patterns of venom variation, in this species.”

line 352. The supplementary videos show the use of tweezers in provoking the bugs which do not seem "featherlight".

Featherlight tweezers were used in general, but not in this demonstration. To avoid confusion, we have removed reference to the type of tweezers used (revised manuscript line 414).

line 243-263. Microinjections into living insects is not trivial, and different methods will greatly impact the results. Please explain in details what kind of needles were used in injecting insects, how far were they introduced, and how were small injection volumes controlled, particularly in the nanoliter scale. Irreproducible microinjection methodology is a major issue in bioassays descriptions with insects, e.g. widespread use of nanoinjection systems such as FemtoJet which do not really control injected volumes.

We have extensive experience in insect injection assays and have performed them using a variety of insects over the past 20 years. In this study, the smallest volume injected was 1.7 microlitres. To assay smaller quantities of venom, dilution series were made in water and the 'venom equivalent' dose reported. To clarify this point, and more specifically describe the equipment and procedure used for microinjections, we re-wrote the "Toxicological assays" paragraph of the Methods Section (revised manuscript lines 529–539):

"Venom toxicity was evaluated by injecting blowflies (*L. cuprina*) or crickets (*A. domesticus*) with gland extracts or venom using a 27G needle on a 1 ml syringe driven by a hand microapplicator (Burkard Scientific) that allows injection of precise microlitre quantities. The needle was allowed to penetrate ~1 mm into the insect. For paralysis and death assays in blowflies, 1.7 μ l of gland protein extract (of a 30 μ l extract of both paired gland compartments of one individual; i.e. 5.6% of the venom in the paired compartments in one individual) was injected into the thorax. Flies were scored for paralysis and death at 15 and 60 min. For escape assays in crickets, 3rd-instar nymphs were injected in the abdomen with 1.7 μ l venom, venom dilution or water. Crickets were then immediately placed upright in the centre of an upturned lid of a 100 mm petri dish, and the time taken to exit the dish recorded."

line 429-431. I do not think the criteria for defining the tentative contigs as housekeeping were clear enough to be reproduced. "Proteins were assigned as 'housekeeping' or 'non-housekeeping' depending on the presence of a predicted signal peptide, predicted transmembrane regions, and ER [please also define "ER" here] retention signals." This is a point I'd prefer that the authors make clearer in their manuscript, as it is a recurrent issue accross OMICS analyses. Objectively the term housekeeping gene can be used to those genes which are expressed at a fairly constant rate accross different cell types, meaning the main function of their phenotype is regulating homeostatis. Therefore objectively assigning 'housekeeping genes' within any OMICS analysis primarily depends on as many tissue/cell-type/organ-specific transcriptomes as possible, usually not available. How does one then differentiate a housekeeping gene from toxin-related genes thus depends on

assumptions that need to be clarified. Specially when minding toxin genes are typically recent duplications of functionally unrelated proteins (e.g. vitellogenin as Api m 12 in bees and OBPs as Sol i 2 and Sol i 4 in fire ants). Some authors deposited microarray comparisons for a few model organisms, while others resorted to counteracting strategies, such as subtractive cDNA libraries from whole bodies or from whole bodies by minus venom glands. The present study cannot rely on comparative expression from other tissues and employed no specific strategy for housekeeping genes. Please thus define more specifically which criteria were used in filtering out housekeeping sequences, and why, based on relevant references. Would the employed method correctly identify, e.g. Apis m 12 among vitellogenin ESTs? This is a fundamental point in venom OMICs studies which is frequently overlooked. Finally I should remark that likely the obtained contigs do not represent complete transcripts, thus an absence of e.g. a signal peptide does not necessarily mean there is none in the actual transcript.

We agree that this is a general issue for OMICs, and venom studies in particular. In our case, we have classified proteins in this way not because it is our primary way to identify venom proteins, but because it makes no sense to include proteins that are very likely to have house-keeping roles in the venom gland, e.g. actin or ribosomal proteins, in the ensuing description of the results. For this reason, we have re-written this section to clarify that we have *excluded* some proteins likely to represent house-keeping proteins from the analysis, and analysed the remainder, which we call “possible venom proteins”. This also allows a sensible normalisation of the results. In response to Eduardo’s question, Api m 12, as well as related non-venom vitellogenins such as NP_001295471.1 (*Athalia rosae*), would be classified as ‘possible venom proteins’ according to this analysis. We have nowhere reported proteins identified only from gland extracts as venom proteins. Moreover, our classification scheme is vindicated by the observation that 95% of ‘possible venom proteins’ detected in the PMG and 78% in the AMG are unambiguously detected *in venom* by MS/MS by the end of the study, whereas no ‘putative housekeeping’ proteins were identified in the venom. To make these points clearer to the reader, we have re-written part of the Methods Section (revised manuscript lines 503–513):

“Although it is not possible to reliably distinguish venom proteins from glandular ‘house-keeping’ proteins using sequence features alone, we nevertheless assigned proteins ‘putative housekeeping’ or ‘possible venom/putative non-housekeeping’ status, in order to focus on likely venom proteins and allow normalisation of quantification data. Proteins were classified as possible venom proteins if they possessed a secretion signal peptide (SignalP D-score >0.4), lacked a C-terminal endoplasmic reticulum KDEL or HDEL sequence, lacked transmembrane regions predicted by TMHMM, and lacked homology ($E < 0.01$) to insect cuticular proteins and collagen; or if they were >95% identical to proteins detected in venom by mass spectrometry. Other proteins were classified as putative housekeeping proteins.”

line 410. I miss a description of details on how the contigs were annotated, including search database and algorithm settings, in a way which would enable reproducing the steps later or in related projects.

Because a similar point was made concerning protein sequence annotation and assignment into groups, we collected this information into a new subsection of the Methods, 'Sequence annotation', that appears on lines 475–485 of the revised manuscript.

line 455 and throughout. No software is declared for the statistical analysis; please clarify. It is known that different programs might affect analyses results. Moreover, I strongly suggest using open sourced software such as R which directly enable including analytical and plot scripts as a supplementary text file. Such relatively simple addition will greatly improve transparency and add value to published data which can be readily reevaluated. For instance, one could easily check whether conclusions would be affected if data had been e.g. square-root-transformed enabling the use of parametric tests, which might have been the preferred route of some other authors. Based on the appearance of the plots perhaps R was already employed, meaning the scripts might be already available.

We agree R would have been a superior choice for statistical analyses and in the future we will endeavor to follow this suggestion. However, in this study we performed statistical analysis using Microsoft Excel with the 'Real Statistics' Add-in. We have now inserted a sentence at the opening of the Statistics subsection of the Methods section to indicate this (revised manuscript line 542). In the interests of transparency, we have made the numerical data for statistical tests available in Supplementary Data S1 and S4.

Finally I am thankful for the opportunity to contribute on an interesting research report. I hope my suggestions and questions contribute to some final version of the present manuscript. I remain open to any communication with the authors on the topic, as relates with my line of research.

We would like to thank Eduardo for his extensive and insightful review and his open attitude that has helped us to significantly improve this manuscript.

Reviewer #3 (Remarks to the Author):

This communication follows previous work of these authors on tissue specialized venom production in various toxic animals. The main point, besides immense information on new proteinaceous compounds, is outlined in the title "The assassin bug *Pristhesancus plagipennis* produces distinct predatory and defensive venoms in separate gland lumens". This observation extends previous work indicating that selective pressures drove the speciation of cone snail venomous duct tissue into two functionally distinct regions (Dutertre et al 2014 Nature Commun. 5; ref 7). In the present paper, the posterior venomous gland of the bug produces a toxin mixture used mostly for hunting, whereas the anterior venomous gland produces mainly a protein mix for defense (paralytic and pain inducing compounds). The data, based on protein chemistry, transcriptomic, anatomic, physiologic and toxinologic analyses, is highly convincing and was carried out very carefully and thoughtfully. It should be mentioned, however, that the findings of cell specialized in secretion of different components in endocrine glands of animals is not new. For example, distinct venomous secretory cells beside nematocysts that produce toxins for predation (usually in tentacles) and others for defense (at the body wall) have been shown in sea anemones (Moran et al 2013 Mar Biotechnol 15, 329). Scorpions have also been described to inject venom of various composition upon stinging as a warning defensive signal versus toxins used for predation (Inceoglu et al 2003 Proc Natl Acad Sci USA 100, 922); Various toxins that appear in organs other than the venom gland were shown by various researchers indicating specialization of secretory cells and localization at various regions of cone snails (Bia et al, 2009 J Proteomics 72, 210; Biggs et al 2008 Toxicon 52, 101; Safavi-Hemami et al 2011 J Biol Chem 286, 22546). Thus, the main contribution of the present paper is the 'state of art' experimental approach showing specialization of two distinct venom glands, which secrete upon different stimuli. Defining of the nerve circuit that controls the muscles of each gland might be a great achievement, and should be considered in future work. Besides the data, the paper is well written and organized including the graphic arts, supporting data and accompanying videos.

We thank Reviewer 3 for their comments. In response to comments about previous related work we have added two sentences to the first paragraph of the Introduction (revised manuscript lines 59–64:

“In cnidarians such as sea anemones, toxin delivery systems are particularly complex and comprise large numbers of individual nematocysts and gland cells across the surface of the animal. Although it is likely that venom delivered by feeding tentacles differs from venom delivered by tissues specialised for defense or intraspecific competition⁸⁻¹⁰, this has not been conclusively demonstrated.”

The inserted references in this passage are:

Reference 8: Macrander, J. *et al.* Tissue-specific venom composition and differential gene expression in sea anemones. *Genome Biol. Evol.* **8**, 2358–2375 (2016).

Reference 9: Mitchell, M. L. *et al.* The use of imaging mass spectrometry to study peptide toxin distribution in Australian sea anemones. *Aust. J. Chem.* **70**, 1235–1237 (2017).

Reference 10: Moran, Y. *et al.* Neurotoxin localization to ectodermal gland cells uncovers an alternative mechanism of venom delivery in sea anemones. *Proc. R. Soc. Lond. B: Biol. Sci.* **279**, 1351-1358 (2012).

We have also referenced and described the work of Incogeu *et al.* 2003 PNAS 100, 922 (revised manuscript lines 57–59).

In summary, we have addressed all of the concerns raised by the three reviewers and we hope that you will find the revised manuscript suitable for publication in *Nature Communications*.

REVIEWERS' COMMENTS:

Reviewer #1 (Remarks to the Author):

The authors did a good job of addressing my remarks, and have added a significant amount of additional information, especially regarding variability within organisms. My only recommendation at this stage is to add the word "pools" in line 160 (...in the AMG and PMG pools).

Otherwise I think this is a very nice and important study and I recommend it for publication.

Daniel Sher

Reviewer #2 (Remarks to the Author):

Dear authors,

I've read through the response to reviews and the new manuscript version, and appreciate how all points raised by reviewers were directly addressed. All my suggestions were provided as originally and I have none to add at this point. Congratulations on the revised version.

My best regards,

Eduardo

NCOMMS-17-18024
Second revisions
12th December 2017

The specific comments of the reviewers and our responses are as follows:

Reviewer #1 (Remarks to the Author):

The authors did a good job of addressing my remarks, and have added a significant amount of additional information, especially regarding variability within organisms. My only recommendation at this stage is to add the word "pools" in line 160 (...in the AMG and PMG pools).

These samples are now referred to as 'pools' for the AMG, PMG and AG (new manuscript line references 163-167).

Otherwise I think this is a very nice and important study and I recommend it for publication.

Daniel Sher

Reviewer #2 (Remarks to the Author):

Dear authors,

I've read through the response to reviews and the new manuscript version, and appreciate how all points raised by reviewers were directly addressed. All my suggestions were provided as originally and I have none to add at this point. Congratulations on the revised version.

My best regards,

Eduardo